# Enhancing Photosynthesis and Plant Productivity through Genetic Modification

**DOI:** 10.3390/cells13161319

**Published:** 2024-08-07

**Authors:** Mansoureh Nazari, Mojtaba Kordrostami, Ali Akbar Ghasemi-Soloklui, Julian J. Eaton-Rye, Pavel Pashkovskiy, Vladimir Kuznetsov, Suleyman I. Allakhverdiev

**Affiliations:** 1Department of Horticultural Science, Faculty of Agriculture, Ferdowsi University of Mashhad, Mashhad 91779-48974, Iran; m_nazari67@alumni.ut.ac.ir; 2Nuclear Agriculture Research School, Nuclear Science and Technology Research Institute (NSTRI), Karaj 31485-498, Iran; akghasemi@aeoi.org.ir; 3Department of Biochemistry, University of Otago, P.O. Box 56, Dunedin 9054, New Zealand; julian.eaton-rye@otago.ac.nz; 4K.A. Timiryazev Institute of Plant Physiology, RAS, Botanicheskaya St. 35, Moscow 127276, Russia; pashkovskiy.pavel@gmail.com (P.P.); vlkuzn@mail.ru (V.K.); 5Faculty of Engineering and Natural Sciences, Bahcesehir University, 34349 Istanbul, Turkey

**Keywords:** genetic manipulation, photosynthetic efficiency, crop productivity, RubisCO modifications, transgenic strategies, microRNA, transcription factors

## Abstract

Enhancing crop photosynthesis through genetic engineering technologies offers numerous opportunities to increase plant productivity. Key approaches include optimizing light utilization, increasing cytochrome *b_6_f* complex levels, and improving carbon fixation. Modifications to Rubisco and the photosynthetic electron transport chain are central to these strategies. Introducing alternative photorespiratory pathways and enhancing carbonic anhydrase activity can further increase the internal CO_2_ concentration, thereby improving photosynthetic efficiency. The efficient translocation of photosynthetically produced sugars, which are managed by sucrose transporters, is also critical for plant growth. Additionally, incorporating genes from C_4_ plants, such as phosphoenolpyruvate carboxylase and NADP-malic enzymes, enhances the CO_2_ concentration around Rubisco, reducing photorespiration. Targeting microRNAs and transcription factors is vital for increasing photosynthesis and plant productivity, especially under stress conditions. This review highlights potential biological targets, the genetic modifications of which are aimed at improving photosynthesis and increasing plant productivity, thereby determining key areas for future research and development.

## 1. Introduction

Photosynthesis utilizes sunlight and water to convert carbon dioxide (CO_2_) into organic compounds. This process can be divided into light-dependent and light-independent reactions. Light-dependent reactions occur in thylakoid membranes, where the electron transfer chain (ETC) operates through protein complexes. Light-independent reactions, also known as the photosynthetic carbon reduction cycle or the Calvin cycle, take place in the stroma of the chloroplast and result in the fixation of CO_2_; these reactions include light-dependent reactions via thioredoxins [1,2]. Photosystem II (PSII), the cytochrome *b_6_f* complex (Cyt *b_6_f*), photosystem I (PSI), and the free electron carriers plastoquinone and plastocyanin together form the photosynthetic electron transport chain that converts free and abundant solar energy into ATP and NADPH, which is essential for the Calvin cycle for CO_2_ fixation [3,4].

Theoretically, most photosynthesis-related enzymes and electron transport components can be targets for improving photosynthesis and plant productivity. However, there are already developments with some of the most promising genes. One of these enzymes is chloroplast ATP synthase, which plays a crucial role in the production of ATP during the light-dependent reactions of photosynthesis. Increasing the expression or activity of ATP synthase can improve the energy efficiency of photosynthesis. Transgenic plants overexpressing the γ-subunit of ATP synthase exhibit increased ATP production, leading to an increased photosynthetic rate and growth [5]. This modification aims to ensure that the energy supply matches the increased demand created by other enhancements in the photosynthetic pathway.

The cytochrome *b_6_f* complex is a crucial component of the photosynthetic electron transport chain and plays a key role in proton translocation and ATP synthesis. Enhancing the expression of cytochrome *b_6_f* subunits can lead to increased electron transport rates and ATP production. Transgenic *Arabidopsis thaliana* plants overexpressing the Rieske FeS protein, a subunit of the cytochrome *b_6_f* complex, exhibit increased photosynthetic efficiency and growth under high light conditions [6]. This modification can help optimize the energy conversion process in plants.

Another important component is ferredoxin-NADP^+^ reductase (FNR). FNR is an enzyme involved in the final step of the electron transport chain in chloroplasts, facilitating the production of NADPH. Overexpressing FNR in transgenic plants has been shown to increase the efficiency of electron transport, leading to increased NADPH availability for the Calvin cycle and other biosynthetic pathways. For example, transgenic *Nicotiana tabacum* plants overexpressing FNR exhibit increased photosynthetic capacity and growth under various light conditions [7]. This modification can improve the overall redox balance and metabolic efficiency of plants.

Stomatal conductance plays a critical role in photosynthesis by regulating the exchange of gases, primarily CO_2_ and water vapor, between the plant and its environment. Stomatal conductance is influenced by the density and opening size of the stomata. One approach to increasing stomatal conductance involves genetic manipulation of key regulatory genes involved in stomatal development and function, such as epidermal patterning factor (EPF) [8] and slow anion channel-associated 1 (SLAC1) [9]. For example, the overexpression of *EPF* can lead to an increased number of stomata, whereas the modification of *SLAC1* can result in stomata that remain open for longer periods, thereby allowing more CO_2_ to enter the leaf [8,10].

Mesophyll conductance is another critical factor influencing photosynthetic efficiency and involves the movement of CO_2_ from the intercellular air spaces within the leaf to the chloroplasts where photosynthesis occurs. An increase in mesophyll conductance can significantly increase photosynthetic capacity, and genetic modification is a powerful tool for achieving this goal. Two key components of this process are aquaporins and carbonic anhydrase, both of which facilitate the diffusion of CO_2_ through the mesophyll [11,12].

Aquaporins are membrane proteins that form channels facilitating the movement of water and small solutes, including CO_2_, across cell membranes. By increasing the expression of specific aquaporins, the diffusion rate of CO_2_ within the leaf mesophyll can be increased, thus accelerating its delivery to the chloroplasts. Studies have shown that plants with relatively high aquaporin activity exhibit increased photosynthetic rates, particularly under conditions in which internal CO_2_ diffusion is a limiting factor [12,13]. Increasing the expression of aquaporins can improve water-use efficiency and photosynthetic performance, especially under stress conditions, such as drought. Transgenic *Oryza sativa* plants overexpressing the aquaporin gene *OsPIP1;1* showed improved water uptake, photosynthesis, and yield under water-limited conditions [13]. This strategy aims to increase the ability of plants to maintain their photosynthetic efficiency during periods of water stress.

Carbonic anhydrase is an enzyme that catalyzes the rapid interconversion of CO_2_ and bicarbonate. By increasing the expression of carbonic anhydrase, the availability of CO_2_ in mesophyll cells can be increased, further promoting efficient photosynthesis. Genetic modifications that upregulate carbonic anhydrase can lead to a more rapid conversion of bicarbonate to CO_2_, ensuring a steady supply of CO_2_ for the photosynthetic machinery [14,15]. This approach is particularly advantageous in environments where CO_2_ availability within the leaf is constrained, such as during high rates of photosynthesis or in densely packed foliage [15].

Transcription factors (TFs) are proteins that regulate gene expression by binding to specific DNA sequences, playing essential roles in photosynthesis-related gene expression and in improving photosynthetic performance. Golden2-like (GLK) TFs are key regulators of chloroplast development and function, with overexpression of GLK1 and GLK2 in *A. thaliana* and *O. sativa* increasing chloroplast biogenesis, increasing chlorophyll content, and improving photosynthetic efficiency [16]. NAC TFs, such as VASCULAR-RELATED NAC-DOMAIN 7 (VND7), increase photosynthetic efficiency by regulating genes involved in chloroplast function and stress tolerance, with overexpression in *A. thaliana* improving photosynthetic performance and biomass production under stress conditions [17]. The AP2/EREBP TF family regulates plant development and stress responses, with the overexpression of certain members, such as dehydration-responsive element-binding protein (DREB), improving the photosynthetic efficiency and drought tolerance of transgenic plants [18]. Basic leucine zipper (bZIP) TFs, including elongated hypocotyl 5 (HY5), increase photosynthetic efficiency and biomass production by promoting light-responsive genes and chlorophyll biosynthesis [19].

Improved plant productivity can occur through increased efficiency of photosynthesis-related enzymes, efficient water use, and regulatory components such as microRNAs (miRNAs). miRNAs, small noncoding RNA molecules, posttranscriptionally regulate gene expression and play crucial roles in various plant processes, including photosynthesis. The overexpression of miR156 and miR396 in *A. thaliana* and *O. sativa* has been shown to increase plant biomass production and improve photosynthetic efficiency by modulating key genes involved in leaf development, chloroplast biogenesis, and cell proliferation [20,21].

This review aims to analyze potential targets, the genetic modifications that lead to improved photosynthetic activity and productivity of plants, as well as the current state of affairs in this area. We conclude that the range of genes and associated mechanisms is quite broad. It not only includes photosynthesis-related enzymes directly related to assimilation and key components of photosystems but also addresses the potential for improved chloroplast assimilate efflux and more efficient use of water and energy through C_4_ enzymes. The Calvin cycle involves many genes that are valuable for genetic modification. Additionally, noncoding miRNAs and transcription factors can accelerate cell division, facilitate transitions to necessary ontogenetic stages, and significantly increase stress resistance.

## 2. Chloroplast Component Modifications

### 2.1. Optimization of Light Harvesting and Pigment Content

Photosynthetic pigments are vital components of the photosynthetic apparatus in all plants, and they exist in various types and at various abundances throughout the photosynthetic apparatus. The main photosynthetic pigments in plants, algae, and cyanobacteria are chlorophylls, carotenoids, and phycobilins, which absorb light [22]. The chlorophyll family absorbs light at different wavelengths and has an important function in enabling photosynthetic organisms to adapt to a variety of different light environments. Carotenoids constitute another important class of plant pigments that have different functions in photosynthesis, such as light absorption, and act as important antioxidants to decrease light damage [23]. In the maritime region, another group of pigments is phycobilins, which allow cyanobacteria and red algae to harvest a broad range of light and live in deep water. Therefore, one way to change the photosynthetic machinery is to increase light harvesting via pigments, increase light protection, and improve photosynthetic efficiency to improve product performance [22].

All carotenoid biosynthesis enzymes are encoded in the nucleus and are transferred to plastids after translation [24]. Phytoene synthase (PSY) is an enzyme that catalyzes the first possible rate-limiting reaction of carotenoid biosynthesis. Lycopene epsilon cyclase (LCYE) catalyzes the synthesis of alpha-carotene and lutein, and ultimately, the production of β-carotene and various xanthophylls, which are dependent on NADPH, is catalyzed by lycopene beta cyclase (LCYB) [25]. The expression of *LCYB* in different plants has relatively specific effects on increasing the synthesis of carotenoids, which not only perform their main function but also protect against various stress factors, including excess light [26]. The insertion of *lycopene beta cyclase* (*LCYB1*) from carrot plants into *N. tabacum* increased the accumulation of carotenoids in transgenic plants, and the plant architecture changed, resulting in longer internodes, faster plant growth, earlier flowering, and increased biomass. In the *DcLCYB1* transgenic lines, the transcript levels of the main genes related to the biosynthesis of carotenoids, chlorophylls, gibberellic acid (GA), and abscisic acid (ABA), as well as genes involved in photosynthesis, significantly increased, increasing the carotenoid, chlorophyll, ABA, and GA contents. The presence of longer internodes causes mature leaves to receive more light and delays leaf senescence, while maintaining greater photosynthetic ability in mature leaves [27].

In general, plants compete with each other for survival, and successful competition requires the absorption of sunlight, even to the extent that this prevents the absorption of light by competing plants [28]. The upper layers of plant canopies absorb more sunlight than is needed for photosynthesis [29]. Therefore, reducing the excessive absorption of sunlight by minimizing or shortening the size of chlorophyll antennae in photosystems may modify photosynthetic solar energy performance and increase plant yield in high-density foliage. For example, compared with the wild type, an *N. tabacum* mutant with a reduced antenna accumulates 25% more biomass in the stems and leaves [30]. In *O. sativa*, the small size of the antenna of the mutants prevented excessive absorption of light and increased the efficiency of PSII, which led to an increase in the rate of electron transfer [31].

Another approach to improve the use of light energy is the possibility of adding light-harvesting procedures that are naturally absent in plants. For photosynthesis, plant pigments only use wavelengths from 400 to 700 nm, which is less than half of the solar spectrum [32]. Some cyanobacteria also contain chlorophyll *d* and *f*, which absorb light over a wide range. Therefore, by inserting cyanobacterial genes that encode the enzymes responsible for the synthesis of chlorophyll *d* or chlorophyll *f*, red light absorption can be extended in plants [33]. When infrared energy is available, photosystems based on chlorophyll *a* are remodeled, and infrared radiation can be absorbed by chlorophyll *f* with lower energy intensity [34].

One way to improve light absorption by plants is to manipulate leaf surface, orientation, and light reflection [35]. Sunlight penetration into the canopy improved with erect leaves in the upper parts of the plants, and in this situation, an increase in the CO_2_ absorption rate was observed [36]. In another study, an *O. sativa dwarf4-1* mutant with an erect leaf phenotype exhibited increased grain performance [37]. Additionally, the *narrow leaf1* (*NAL1*) gene in *O. sativa*, which regulates leaf blade morphology and leaf vein pattern, has pleiotropic impacts on leaf anatomy, altering the rate of photosynthesis on the leaf surface and grain yield [38]. A partially functional *NAL1* allele helps balance leaf photosynthesis and plant architecture in the field, as well as increase biomass and seed productivity in *O. sativa* [39].

### 2.2. Photosystem II

Under field conditions, one of the main constraints on photosynthesis is light inhibition. When light conditions are optimal for plants, the light absorbed by PSII is used for photochemical reactions, such as ETC and CO_2_ fixation [40]. However, under excessive light conditions, plants must protect their photosynthetic machinery from light damage. Plants regularly dissipate excess radiation absorbed by photosystems through nonphotochemical quenching (NPQ) mechanisms and, in this way, protect the photosynthetic apparatus [41]. PSII is essential for the light-dependent reactions of photosynthesis, in which water is split to release oxygen and electrons. Increasing the stability and efficiency of PSII components can lead to improved overall photosynthetic performance. Transgenic approaches have focused on overexpressing the D1 protein of PSII, which plays a crucial role in the repair cycle of the photosystem. For example, overexpression of the D1 protein in *A. thaliana* led to increased tolerance to photooxidative stress and increased photosynthetic rates [42].

For NPQ, under high light conditions, conformational alternations in the PsbS subunit of PSII and the function of the xanthophyll cycle are required for the transformation of violaxanthin to zeaxanthin [41,43]. Zeaxanthin acts as an antioxidant and protects chloroplasts against oxidative light damage [44]. Lutein is another type of xanthophyll that directly quenches excited chlorophyll [45]. The compact structures of pigment-protein complexes limit the interaction of zeaxanthin with chlorophylls at potential excitation quenching sites. PsbS interferes with compact clusters between thylakoid membrane proteins, thereby enabling ordinary interchange and incorporation of xanthophyll cycle pigments into such structures [46].

Compared with those in wild-type plants, canopy radiation utilization performance and seed productivity under high light conditions were greater in PsbS-overexpressing plants [47]. PsbS is involved in the NPQ mechanism by directly affecting the rate at which the excitation energy absorbed by PSII is wasted as heat. Photosynthetic activity, including the functions of PSII and plastoquinone A, contributes to the signal needed for stomatal opening in response to light. Moreover, excess light energy is dissipated as heat through NPQ, a process regulated by PsbS. This gene is potentially vital for water-use efficiency in plants. The overexpression of PsbS increased water-use efficiency in transgenic *N. tabacum* before flowering [48].

The combined overexpression of *violaxanthin de-epoxidase (VDE)*, *PsbS*, and *zeaxanthin epoxidase (ZEP)* in *N. tabacum* induced photoprotection against high light and increased photosynthetic efficiency. The expression of the same three genes induced photoprotection in *A. thaliana,* but reduced the growth of transgenic plants. In *S. tuberosum* plants expressing *VDE*, *PsbS,* and *ZEP*, similar to *N. tabacum* and *A. thaliana* plants, NPQ was induced, but no overall increase in the photosynthetic rate or growth rate was observed in the transgenic plants. The lack of benefit from enhanced photoprotection could be due to decreased light utilization efficiency under high light conditions, a consequence of the overly strong induction of NPQ [49].

### 2.3. C. cytochrome b_6_f Complex

The cytochrome *b*_6_*f* complex is important for photosynthetic regulation because it catalyzes the rate-limiting step in thylakoid electron transfer [50]. In the Cyt *b*_6_*f* homodimer, each monomer is composed of eight subunits: the major subunits, RieskeFeS protein, cytochrome *b*_6_ (PetB), cytochrome *f* (PetA), and subunit IV (PetD), and the minor subunits, PetG, PetL, PetM, and PetN [51] (Figure 1). The level of Rieske protein can regulate the abundance of Cyt *b*_6_*f,* along with that of PetM [6,52,53]. The overexpression of the Rieske protein in *A. thaliana* resulted in increases in the electron transport chain and the CO_2_ assimilation rate, along with a reduction in NPQ. This overexpression also led to an increase in other subunits of the Cyt *b*_6_*f* complex [6]. Additionally, in transgenic *N. tabacum* plants, the rate of electron transfer through the electron transport chain and the rate of CO_2_ fixation are simultaneously determined by Cyt *b*_6_*f* [54]. The overexpression of the RieskeFeS protein in *Setaria viridis* increased the content of Cyt *b*_6_*f* in mesophyll and bundle sheath cells, whereas no change in the abundance of other photosynthetic proteins was detected. In addition, the light conversion efficiency of the transgenic plants was improved in both photosystems, and a stronger proton motive force was generated across the thylakoid membrane. This increased proton motive force could increase the rate of CO_2_ absorption from the environment. Therefore, removing the limitations of electron transfer can increase C_4_ photosynthesis [52]. The overexpression of the RieskeFeS protein in *N. tabacum* resulted in a 40% increase in the abundance of functional Cyt *b*_6_*f*. In these transgenic plants, the establishment of a proton gradient across the membrane was accelerated, and the activity of Cyt *b*_6_*f* and the capacity for electron transfer increased. However, under in vitro or field conditions, no substantial increase in steady-state electron transfer rates or CO_2_ assimilation was observed in plants overexpressing the RieskeFeS protein. These findings suggest that the in vivo function of this complex may only temporarily increase with changes in radiation [50].

In *Sorghum bicolor*, transgenic plants overexpressing the RieskeFeS subunit did not show significant variation in the levels of other photosynthetic proteins or chlorophyll. However, these plants exhibited improved PSII performance, a greater rate of CO_2_ assimilation, and a faster NPQ reaction. Considering that the steady-state electron transfer rate and CO_2_ assimilation in the transgenic plants were similar to those in the control plants, it was concluded that in addition to Cyt *b*_6_*f*, there are other limiting factors in the electron transfer pathway under high light and high CO_2_ conditions. Increased photosynthesis leads to increased biomass and grain yield in transgenic plants [55].

### 2.4. Photosystem I

Ferredoxin (Fd) is an electron transport protein that belongs to a family of low-molecular-weight proteins with a simple [2Fe-2S] cluster attached to four conserved cysteine residues [56]. In the ETC, Fd is located on the stromal side of the thylakoid membrane and transports electrons from PSI to a broad range of enzymes such as ferredoxin—NADP(+) reductase (FNR), ferredoxin-thioredoxin reductase (FTR), nitrite reductase (NiR), glutamine-oxoglutarate aminotransferase (GOGAT), sulfite reductase (SiR), fatty acid desaturase (FAD), and chlorophyll *a* oxygenase (CAO) [57].

Ferredoxin-NADP+ reductase (FNR) is an enzyme involved in the final step of the electron transport chain in chloroplasts, facilitating the production of NADPH (Figure 1). Overexpressing FNR in transgenic plants has been shown to increase the efficiency of electron transport, leading to increased NADPH availability for the Calvin cycle and other biosynthetic pathways. For example, transgenic *N. tabacum* plants overexpressing FNR exhibit increased photosynthetic capacity and growth under various light conditions [7]. This approach can improve the overall redox balance and metabolic efficiency of plants.

Fd proteins (FdC1 and FdC2) are encoded by the nuclear genome. FdC1 and FdC2 have extended regions at the C-terminus of the 2Fe-2S cluster. Mutation or overexpression of *FdC* genes changed photosynthetic electron transport chain rates in *O. sativa* and *A. thaliana* plants [58,59]. There is one copy of each *FdC* gene in the *Zea mays* genome. In both mesophyll cells and bundle sheath cells, the *ZmFdC2* gene was detected. Mutation of this gene interferes with the photosynthetic electron transport chain and causes chloroplast collapse, which can cause *Z. mays* death. The *ZmFdC2* gene is specifically expressed in photosynthetic tissues and has a specialized function in photosynthesis. Light treatment induces this gene, and the proteins encoded by this gene are localized in the chloroplast [59,60]. In addition to the role of Fd in many biological processes, including chlorophyll metabolism, carbon fixation, nitrogen assimilation, and fatty acid synthesis, Fd has been demonstrated to play a role in stress tolerance [61]. The tolerance of tomato plants overexpressing *pFd08* to high-temperature stress was markedly increased. Additionally, the calli of the *PpFd08* transgenic apple plants exhibited thermal resistance, possibly due to the modulation of the expression of genes involved in heat tolerance due to the expression of the *PpFd08* gene [62].

Under various environmental stresses, Fd transcript and protein levels decrease [63]. In cyanobacteria and some algae, there is a compensatory system for this reduction. In this system, electron transport continues with the inducible expression of flavodoxin (Fld), an isofunctional 19 kDa electron carrier containing flavin mononucleotides [64]. In fact, under stress conditions, Fld replaces ferredoxin, functions, and prevents disruption of the electron transfer chain [65]. Flavodoxin is not encoded in plant genomes. However, cyanobacteria and some algae harbor this system, which shows that this adaptation system was removed from the genomes of higher plants during evolution [66]. Many studies have shown that the expression of the flavodoxin gene improves the photosynthetic activities of plants and, as a result, creates tolerance to various stressors. The expression of plastid-targeted cyanobacterial *Fld* in *Solanum lycopersicum* cv. Moneymaker led to increased pigment content and photosynthetic activity in each leaf section. Additionally, lines expressing flavodoxin had a greater harvest index because of the production of more fruits per plant [67].

In another study, compared with wild-type plants, transgenic *N. tabacum* plants overexpressing *Fld* and *betaine aldehyde dehydrogenase* (*BADH*) presented improved quantum yields. These researchers also reported that the activity of antioxidants in transgenic lines increased, and the amounts of malondialdehyde (MDA) and hydrogen peroxide (H_2_O_2_) decreased in these lines. These researchers reported that reducing the amount of ROS generated in the photosynthetic electron transport chain and motivating the function of antioxidant enzymes modified the tolerance of transgenic plants to cadmium toxicity [68]. Additionally, in another experiment, transgenic lines of the drought-sensitive Persian walnut tree harboring the *Fld* gene were found to be able to tolerate high levels of drought stress (10 and 12% PEG) in vitro [69].

### 2.5. Electron Transport Chain

Light-initiated charge separation occurs at the photosystem II (PSII) and photosystem I (PSI) levels. This process begins with the oxidation of water in the PSII complex, where electrons are extracted from water molecules. These electrons are subsequently transferred through a series of carriers, including plastoquinone, cytochrome *b_6_f*, and plastocyanin, ultimately reaching PSI. Concurrently, protons are translocated to the thylakoid lumen, creating a proton gradient that drives ATP synthesis via ATP synthase (Figure 1). The NADPH produced at the end of the chain, together with ATP, is used for CO_2_ absorption [4]. Specifically, the genetic systems in the nucleus and chloroplasts of plants synthesize the pigment and redox cofactors PSII, PSI, and Cyt *b*_6_*f* [70,71].

Yamori and Shikanai (2016) provided a comprehensive review of the critical roles played by cyclic electron transport (CET) around photosystem I (PSI) in plants [72]. This process is pivotal for the stabilization and optimization of photosynthesis, particularly under fluctuating environmental conditions. CET contributes to the generation of a proton gradient across the thylakoid membrane, which is essential for ATP synthesis, thereby supporting the Calvin cycle and other energy-demanding processes within chloroplasts. The authors highlight the adaptive advantages of CET, allowing plants to manage the balance between ATP and NADPH production and mitigating the formation of reactive oxygen species (ROS), which can cause cellular damage. This review underscores the integral role of CET in enhancing plant growth and productivity, especially under stress conditions such as drought or high light intensity [72,73].

In accordance with the findings of Yamori and Shikanai, recent studies have further elucidated the mechanisms and benefits of increased PSI activity in plants [72]. For example, Basso et al. (2022) investigated the role of flavodiiron proteins (FDPs) in *A. thaliana* under fluctuating light conditions [74]. Their study demonstrated that FDPs can significantly increase the rate of CO_2_ assimilation. The key genes studied in this work included *FLV1* and *FLV3*, which encode the flavodiiron proteins critical for this process. This enhancement occurs by facilitating alternative electron flow, which helps to quickly dissipate excess energy and reduce the buildup of ROS during sudden changes in light intensity. As a result, plants with elevated FDP levels show improved photosynthetic efficiency and improved growth performance in environments with variable light conditions, reflecting the practical application of modulating PSI activity for agricultural productivity. Similarly, the research by Wada et al. (2018) provides additional insights into the functionality of FDPs in *O. sativa* [75]. They explore how FDPs can replace CET around PSI without negatively impacting CO_2_ assimilation. The primary genes analyzed in their study were *FLV1* and *FLV4*, which are responsible for encoding the essential flavodiiron proteins in *O. sativa*. This study revealed that FDPs can effectively support the photosynthetic machinery by maintaining ATP production and stabilizing the photosynthetic apparatus, even when CET pathways are compromised. This substitution ensures that CO_2_ fixation continues efficiently, thereby supporting overall plant growth and yield. The findings of Wada et al. emphasize the versatility and importance of FDP in maintaining robust photosynthetic performance and plant resilience [75].

### 2.6. Chloroplast ATP Synthase

Chloroplast ATP synthase is an enzyme complex in the thylakoid membrane of chloroplasts that plays a critical role in the production of ATP during the light-dependent reactions of photosynthesis. This enzyme uses the proton gradient generated by the electron transport chain to synthesize ATP from ADP and inorganic phosphate (Pi). The ATP produced in this way is essential for initiating the Calvin cycle and other cellular processes. Improving the expression or activity of chloroplast ATP synthase may increase the energy efficiency of photosynthesis [76]. By increasing ATP production, plants can better meet the energy demands of improved photosynthetic processes, resulting in improved growth and productivity (Figure 1).

Transgenic *N. tabacum* plants overexpressing the γ subunit of ATP synthase showed increased ATP production, increased photosynthetic rates, and improved growth. This suggests that increased ATP synthase expression may increase the overall energy efficiency of photosynthesis, supporting greater biomass accumulation and productivity [77].

Chloroplast ATP synthase consists of two main components: CF1 (catalytic component), which is located on the stromal side of the thylakoid membrane and contains catalytic sites for ATP synthesis [78]. CF0 (a membrane component) is embedded in the thylakoid membrane and forms a channel through which protons (H+) flow, driving the synthesis of ATP from ADP and Pi [79]. The light-dependent reactions of photosynthesis generate a proton gradient across the thylakoid membrane. Protons return to the stroma through the CF0 channel, driving the rotation of the CF1 catalytic core. With more ATP available, the Calvin cycle can operate more efficiently, resulting in a greater rate of carbon fixation and overall photosynthesis. In *O. sativa*, overexpression of AtpD, the nuclear-encoded subunit of chloroplast ATP synthase, regulates both the redundancy of the complex and ATP synthase activity. As a result, these plants presented a greater rate of CO_2_ assimilation, which was correlated with increasing AtpD content. Compared with those in wild-type plants, ATP production in transgenic plants increased the carboxylation rate and cyclic electron flow [55].

## 3. Carbon Assimilation Efficiency

### 3.1. RuBisCO as a Target to Improve Carbon Assimilation Efficiency

As a key enzyme in photosynthesis, Rubisco has limitations. It is not efficient enough for the current environmental conditions and agricultural activities, but there is sufficient diversity in this enzyme in nature, which shows that there is a possibility of improving the catalytic attributes of Rubisco or its side reactions. Therefore, providing solutions to significantly increase the CO_2_-fixing capability of plants by changing the function of Rubisco (RBC) and/or Rubisco activase (RA) is one of the important aspects of current research [80]. The Calvin cycle includes three main processes, as shown in Figure 2 [81]. RBC has some properties that surprisingly render it inefficient, thereby reducing its photosynthetic efficiency. For example, during photorespiration, much of the energy is lost through the futile reaction of RBC with oxygen, leading to the release of CO_2_, which has already been fixed. Additionally, because RBC is slow, large amounts of RBC are required to support a sufficient photosynthetic rate. Consequently, one of the main targets for engineering to increase photosynthesis is improving the performance of RBC in the carbon fixation cycle [80].

RBC has eight small subunit (RbcS; ~12–18 kDa) protomers and four large subunits (RbcL; ~50–55 kDa) that hold the dimers together. It forms a hexameric L8S8 complex (~530–550 kDa) in the shape of a cylinder with a diameter of approximately 10 nm [82].

C_4_ plants and cyanobacteria have RBC variants with high specificity, and the insertion of these RBC into C_3_ plants can improve the photosynthetic efficiency of crops. Previous studies on the expression of large subunits of *Flaveria bidentis* (C_4_) and *Flaveria pringlei* (C_3_) RBC in transgenic *N. tabacum* plants revealed that the substitution of methionine-309 with isoleucine led to an increase in the RBC carboxylation rate [83,84]. The complete replacement of the *O. sativa RbcS* gene with the *Sorghum bicolor* (C_4_) *RbcS* gene in *O. sativa* resulted in a hybrid RBC enzyme exhibiting properties characteristic of C_4_ plants [85,86]. The insertion of RBC with large and small subunit genes found in cyanobacteria into *N. tabacum* plants led to the production of hybrid RBC transgenic plants. The rate of CO_2_ uptake per Rubisco content increased in the transgenic plants, but their growth was slower than that of the nontransformed plants [87].

In another study, the expression of *RBC* without large subunits from the red alga *Griffithsia monilis* was investigated in *N. tabacum* chloroplasts. The results revealed successful transcription of red alga *RBC* genes in *N. tabacum* chloroplasts; however, the transgenic lines lacked functional L8S8 RBC enzymes and were only able to grow in sucrose-containing media. These researchers [88] concluded that there is an incompatibility of *N. tabacum* chloroplast folding chaperonins with the *G. monilis* large subunit to produce the correctly folded L2 dimer intermediate that leads to the degradation of both the *G. monilis* large and small subunits. Therefore, the coexpression of adaptive chaperones is essential for the successful assembly of red alga RBC in plants [88].

### 3.2. Rubisco Assembly Factors

Rubisco assembly factors (RAFs) are crucial for the proper assembly and function of Rubisco (ribulose-1,5-bisphosphate carboxylase/oxygenase), which is the primary enzyme responsible for carbon fixation during photosynthesis. There are several key aspects of Rubisco assembly factors (Figure 1).

Increasing RBC abundance in C_4_ plant chloroplasts is another way to modify catalytic attributes, which may enhance the function of RBC and increase the carbon uptake rate. RAF is the main intermediate in the regulation of RBC activity and plays a critical role in carbon assimilation and the Calvin cycle [81,89]. In one study, the overexpression of large and small RBC subunits with the RBC assembly chaperone RAF1 led to a 30% increase in RBC content [90]. Recently, this approach was used to investigate the effects of RAF on carbohydrate metabolism and the response to heat stress in the important medicinal plant *Dryopteris fragrans*. The results revealed that heat stress significantly increased the expression of *DfRaf*, leading to increased RBC activity, RCA, and phosphoribulokinase under stress conditions. However, under these conditions, the contents of H_2_O_2_, O_2_^−^, and MDA in the *DfRaf-OV-L2* and *DfRaf-OV-L6* transgenic lines were markedly lower than those in the control plants. Additionally, these lines presented greater contents of photosynthetic pigments, soluble sugars, and proline and greater abilities to scavenge ROS [91].

At high temperatures, two important factors affect RBC: (1) reducing the activation mode of RBC and (2) disrupting the synthesis of the RBC enzyme complex [84,91]. At high temperatures, an increase in the rate of RBC inactivation relative to the rate of RBC activation driven by RCA activity results in RBC inactivation [84,92]. The high sensitivity of Rubisco activase (RCA) to heat inhibits photosynthesis and plant productivity under heat-stress conditions. RCA is a protein that interacts with an RBC to convert it back into its active form after it becomes inactivated. Therefore, increasing the thermal stability of RCA can help maintain photosynthetic activity at high temperatures [93]. By using expression profiling, the reactions of three wheat RCA isoforms, *TaRCA1-β*, *TaRCA2-α*, and *TaRCA2-β*, to heat stress were tested. Like in *Gossypium hirsutum* and *Z. mays*, in response to heat stress, the expression of heat-resistant RCA isoforms, especially the *TaRCA1-β* gene, is induced in wheat regardless of cultivar [94]. In the model species *A. thaliana*, improving the thermal stability of RCA via genetic manipulation increased growth, development, and photosynthesis under high-temperature conditions [95].

Similarly, in *O. sativa*, overexpression of *Z. mays RCA* resulted in improved photosynthetic performance under heat stress conditions [96]. A comparison of RCA-overexpressing *Cucumis sativus* plants with nontransformed plants revealed an increase in the abundance of RBC subunit mRNAs and increases in RBC enzyme and RCA activities in the transgenic plants, which resulted in a better photosynthetic rate and significantly improved thermal tolerance at 40 °C [97].

In a previous study, the overexpression of genes encoding heat-resistant Rubisco activase (RCA) from wild *Oryza australiensis* in domesticated *O. sativa* led to a seed number increase of up to 150% under heat stress conditions. Transgenic plants exposed to 45 °C had greater leaf elongation rates and greater numbers of stems. These results showed that the heat-tolerant form of RCA increased carbohydrate accumulation and storage in *O. sativa* throughout its life cycle, and under heat stress, it significantly improved performance during the vegetative phase [98].

A naturally rather heat-resistant RCA (RCA isoform α gene, LtRCA) from the desert shrub *Larrea tridentate*, which often experiences temperatures above 40–45 °C in summer, was expressed in *A. thaliana*. In transgenic plants, improved photosynthesis was observed, leading to increased seed production under high-temperature conditions. Since abiotic stresses rarely occur alone and often occur in different combinations, especially for drought and heat stresses, this study tested the resistance of these transgenic plants to simultaneous drought and heat stress. The research team inserted the *RCA* gene together with the *AVP1* gene (tolerance to salinity and drought). As expected, the *AVP1/RCA*-overexpressing plants were more tolerant to high temperature, water deficit, salinity, and multiple stresses. Additionally, compared to plants overexpressing *AVP1* or *RCA* alone, plants overexpressing *AVP1/RCA* presented increased plant biomass and grain yield under combined heat and drought stress conditions [93].

*RCA* expression is regulated by the transcription factor Ghd2, which binds to the “CACA” motif in the *RCA* gene promoter via its CCT domain. The *RCA* gene has alternative transcripts: *RCAS*, which is expressed under normal conditions, and *RCAL*, which is expressed under drought stress conditions. Plants overexpressing *RCAL*, similar to plants overexpressing *Ghd2*, were more sensitive to drought stress than were the wild-type plants. Furthermore, the *rcal* mutant plants did not exhibit any yield reduction under normal conditions but presented greater drought tolerance and delayed leaf senescence. Therefore, Ghd2 causes leaf senescence by upregulating the expression of *RCAL* under drought stress, and the *rcal* mutant can be used for breeding programs as a drought-resistant variety [99].

An evaluation of the combined insertion of *RBC* and *RBA* genes in *O. sativa* at 25 and 40 °C revealed that, compared with that in the wild type, the level of RCA was 153% greater in the *oxRCA* (*Rubisco activase*–overexpressing line) and 138% greater in the *oxRCA-RBC* (*Rubisco* and *Rubisco activase*–co-overexpressing line), and the content of RBC in the *oxRCA* plants was 27% lower and similar to that in the *oxRCA-RBC* plants. The CO_2_ assimilation rate of the wild-type plants at 40 °C was lower than that at 25 °C, which was attributed to the inactivation of RBC by heat. Additionally, at 40 °C, the dry weight of *oxRCA-RBC* lines increased by 26% compared to that of the wild type [100]. The benefits of improving the performance of RBC can be summarized as follows (Figure 2) [80].

The findings of Qu et al. (2023) represent a significant advance in the quest to improve plant heat tolerance by modifying RCA in C_3_ plants [101]. In plants, C_3_ RCA is vital for maintaining Rubisco functionality, especially under stress conditions such as high temperatures. This article delves into the genetic basis of heat tolerance by examining two specific genes, *RCA1* and *RCA2*, which encode different isoforms of RCA. These isoforms play distinct roles in the regulation and activation of Rubisco, and their functionality is highly sensitive to temperature fluctuations. The authors highlighted that under high-temperature conditions, the effectiveness of RCA diminishes, resulting in reduced photosynthetic capacity and, consequently, lower yields. To address this issue, they proposed genetic modifications to increase the thermostability and activity of RCA [101]. Their approach involves altering specific amino acids in the RCA protein to improve its heat stability. By employing site-directed mutagenesis and protein engineering techniques, researchers have successfully created modified versions of RCA with increased heat tolerance.

### 3.3. Sedoheptulose-1,7-Bisphosphatase

Sedoheptulose 1,7-bisphosphatase (SBPase) is an essential enzyme in the Calvin cycle and is the major carbon fixation pathway in photosynthesis (Figure 1). SBPase catalyzes the conversion of sedoheptulose-1,7-bisphosphate to sedoheptulose-7-phosphate, which is a crucial step in the regeneration of ribulose-1,5-bisphosphate (RuBP), which accepts CO_2_ in the first step of the Calvin cycle, and its availability directly affects the efficiency of carbon fixation. SBPase is involved in the regeneration phase of RuBP, and its activity ensures the availability of RuBP for the carboxylation process. By increasing SBPase activity, the regenerative capacity of RuBP can be increased, leading to increased Calvin cycle efficiency and overall photosynthetic efficiency. Transgenic *N. tabacum* plants overexpressing SBPase showed significant improvements in photosynthesis. Transgenic *N. tabacum* plants with increased SBPase activity exhibited increased photosynthetic rates, greater biomass accumulation, and improved growth under both ambient and elevated CO_2_ conditions. These results indicate that increasing SBPase activity can significantly improve the overall efficiency of the Calvin cycle. Plants with increased SBPase activity may also exhibit better tolerance to environmental stresses, such as fluctuating CO_2_ levels and other abiotic factors [102,103,104,105].

### 3.4. Genes Involved in C4-Type Photosynthesis

Efforts to increase the rate of photosynthesis by inserting photosynthesis-related enzymes from C_4_ plants into C_3_ plants have shown promise, although the results have thus far been modest. One of the key enzymes involved in this process is phosphoenolpyruvate carboxylase (PEPC), which plays a key role in C_4_ photosynthesis and in crassulacic acid metabolism (CAM), which facilitates the initial fixation of CO_2_ into oxaloacetate. PEPC catalyzes the conversion of phosphoenolpyruvate (PEP) and bicarbonate (HCO_3_^−^) to oxaloacetate, which is then converted to malate [106]. This step is critical for C_4_ plants because it effectively concentrates CO_2_ near Rubisco, thereby increasing carboxylation efficiency and minimizing photorespiration. In plants, CAM-PEPC also plays a vital role in the temporal separation of carbon fixation and carbon assimilation, allowing these plants to conserve water. The insertion of C_4_-specific enzymes, such as PEPC, into C_3_ plants, aims to increase photosynthetic efficiency by increasing the internal CO_2_ concentration around Rubisco. The overexpression of PEPC has been investigated in various C_3_ plants, with promising results.

Transgenic *O. sativa* plants overexpressing the *Z. mays* PEPC gene exhibited increased photosynthetic rates and biomass production under high-CO_2_ conditions. This improvement was due to the increased availability of CO_2_ in the Calvin cycle, which increased the overall efficiency of photosynthesis.

CAM photosynthesis has attracted significant interest because of its water-use efficiency. Approximately 90% of higher plants undergo C_3_ photosynthesis, and approximately 3% of the remaining plants are C_4_ plants. Approximately 6% of CAM plants, similar to C_4_ plants, have evolved several times. C_4_ and CAM plants enrich carbon dioxide molecules around RUBISCO for carbon assimilation reactions, but while C_4_ plants have a CO_2_ concentration mechanism (CCM)that spatially separates the Calvin cycle and electron transport chains of the photosynthetic pathways, CAM plants use CCM to temporarily separate the photosynthetic pathways. The genomes and important genes involved in improving water-use efficiency in CAM plants have been well studied.

For example, a study by VanBuren et al. (2015) sequenced the pineapple genome, revealing the genetic basis of CAM photosynthesis [107]. The authors identified several key genes, including phosphoenolpyruvate carboxylase (PEPC), phosphoenolpyruvate carboxykinase (PEPCK), malate dehydrogenase (MDH), NADP-malic enzyme (NADP-ME), and pyruvate phosphate dikinase (PPDK). These genes are critical for fixing CO_2_ at night and releasing it during the day, allowing efficient water use under dry conditions [107]. Yang et al. (2017) examined the Kalanchoe genome and identified similar key genes involved in CAM photosynthesis. The study highlighted the roles of PEPC, MDH, NADP-ME, starch synthase (SS), starch phosphorylase (SP), and various transport proteins. These results highlight the convergent evolution of CAM, where different plant species have independently evolved similar genetic solutions to thrive in arid environments [108]. Yin et al. (2018) examined the Agave genome and identified key genes and proteins that have undergone daily regulatory changes and positive selection to promote the evolution of CAM photosynthesis [109]. The important genes included PEPC, NAD-malic enzyme (NAD-ME), NADP-malic enzyme (NADP-ME), starch synthase (SS), starch phosphorylase (SP), and circadian clock genes. This study revealed how ancient plant proteins were coopted and selectively enhanced to support CAM, highlighting the adaptive significance of genetic rearrangement in response to environmental pressure [109].

Research with transgenic CAM plants has been carried out; for example, the overexpression of the *PEPC* gene in Agave also significantly improved plant growth [110]. In another study, introducing a transcription factor from the CAM plant Agave into the C_3_ *Populus tremula* plant increased biomass yield by 166% [111]. Because the CAM pathway is a natural plant innovation for improving photosynthesis and productivity in response to lower CO_2_ concentrations and water stress, it is important to add a discussion on CAM photosynthesis. Yang et al. (2015) developed a roadmap for CAM metabolic research to improve sustainable food and bioenergy production in a changing climate [112].

All these studies identified PEPC as a critical enzyme for nocturnal CO_2_ fixation, highlighting its universal importance in CAM photosynthesis. Genes involved in carbohydrate storage and mobilization, such as *starch synthase (SS)* and *starch phosphorylase (SP),* have been identified in several CAM plants. Regulatory networks and circadian clock genes play important roles in optimizing CAM activity according to the circadian cycle.

By optimizing the photosynthetic process, increased CO_2_ fixation by PEPC may also help plants better tolerate environmental stresses, such as high light intensity and temperature fluctuations. Although the overexpression of C_4_ enzymes, such as PEPC, in C_3_ plants has potential, several challenges remain. Ensuring that the insertion of PEPC does not disrupt the overall metabolic balance of the plant is critical. The integration of C_4_ pathways into C_3_ plants requires careful regulation to avoid negative effects on plant growth and development. To fully realize the benefits of C_4_ photosynthesis in C_3_ plants, a combination of genetic modifications may be needed, including the insertion of other C_4_-specific enzymes and regulatory elements [113].

### 3.5. Photorespiratory

Photorespiration is a process dependent on light. In this process, O_2_ is absorbed, and CO_2_ is released at the same time. In C_3_ plants, photorespiration is the second metabolite flux after photosynthesis (Figure 2) [114,115]. In the 1970s and the 1980s, the binary function of RBC was discovered. During the photorespiratory cycle, the reaction of ribulose-1,5-bisphosphate oxygenase leads to the loss of carbon and nitrogen [116]. Under heat and drought stress, the internal CO_2_ concentration of the leaf approaches the apparent CO_2_ compensation point; therefore, photorespiration may further increase.

Since 1900, the atmospheric CO_2_ concentration has increased by 100 ppm, reducing the ratio of photorespiration to photosynthesis in C_3_ plants by approximately 25%. This change will continue to impact photosynthesis and yield in future climates [117,118]. Eliminating photorespiration is predicted to increase photosynthesis by 12–55% [118,119]. Consequently, targeting photorespiration through bioengineering has become a primary focus for improving crop productivity.

Recent efforts have introduced new photorespiration pathways. Pathway 1 involves diverting photorespiratory glycolate to glycerate within the chloroplast, shifting CO_2_ release from mitochondria to chloroplasts, and reducing ammonia release [120]. In 2007, *E. coli* glycolate dehydrogenase (*GDH*), glyoxylate carboligase (*GCL*), and tartronic semialdehyde reductase (*TSR*) were inserted into *A. thaliana* chloroplasts. This pathway directs 75% of glycolate back into the Calvin cycle, thereby increasing net photosynthesis and biomass production in *A. thaliana* [121]. In *O. sativa*, complete insertion (CI) involves five genes encoding *GCL*, *TSR*, and three subunits of *GDH*, whereas partial insertion (PI) involves only the three subunits of *GDH*. Compared with nontransformed plants, both CI and PI plants presented improved photosynthetic performance, biomass, and seed yield. This pathway reduces the oxygenase function of RuBisCO, enhancing the Calvin cycle and increasing carbon sequestration, as indicated by harvest index evaluations [122].

Pathway 2 involves the insertion of two *E. coli* enzymes that convert glyoxylate to hydroxypyruvate in the peroxisome. This pathway shifts CO_2_ release from the mitochondria to peroxisomes, with 1/4 of glycolate carbon released as CO_2_ and 3/4 converted to 3-phosphoglyceric acid. This approach has shown limited success in *N. tabacum* [118,123].

Pathway 3 involves the introduction of two enzymes, malate synthase and malate dehydrogenase, leading to the formation of reducing equivalents and shifting of CO_2_ release reactions from the mitochondria to the chloroplasts. Preliminary results in transgenic *A. thaliana* indicated increased photosynthesis and biomass via this pathway [124,125].

The efficiency of the three alternative photorespiratory pathways was tested in *N. tabacum*. In the first route, five genes from the glycolate oxidation process in *E. coli* were used. In the second pathway, malate synthase, glycolate oxidase from plants, and catalase from *E. coli* were used, whereas in the third pathway, plant malate synthase and glycolate dehydrogenase from green algae were used. The results revealed a 13% increase in biomass under pathway 1. However, pathway 2 did not significantly differ from that of the wild type, but the insertion of pathway 3 increased the biomass in the greenhouse and increased the photosynthetic light utilization performance by 17% in the field [126]. Recently, a novel respiratory pathway, the glycolate oxidation cycle (GOC), was successfully established in *O. sativa* chloroplasts. This pathway is catalyzed by three *O. sativa* enzymes—a glycolate oxidase, an oxalate oxidase, and a catalase—and is characterized by a lack of reducing equivalents during the complete oxidation of glycolate to CO_2_. The transgenic *O. sativa* plants presented increased photosynthetic performance, nitrogen levels, biomass, and seed yield under both greenhouse and field conditions. The GOC plants performed better under light conditions than did the control plants, and this improvement was due mainly to the photosynthetic effect of concentrated CO_2_ rather than an improved energy balance [118].

A photorespiratory bypass known as GCGT (consisting of genes encoding *O. sativa* glycolate oxidase and *E. coli* catalase, glyoxylate carboligase, and tartronate semialdehyde reductase), which increases the concentration of chloroplastic CO_2_ and results in an increased photosynthesis rate and yield, has been tested in *O. sativa* [127]. Compared with those of the wild type, plants transformed via the GOC and GCGT pathways presented greater food quality and cooking quality because the grain protein content increased significantly by 11.27% and 14.97%, respectively, which was probably due to the increase in total nitrogen and transport and limited transfer of carbohydrates in the transgenic plants [128].

## 4. Cellular Transport and Regulation

### 4.1. Sucrose Transporters

The efficient movement of photosynthetically produced sugars is critical for plant growth and development. Sucrose transporters (SUTs) facilitate the loading and unloading of sucrose into phloem cells for distribution throughout the plant (Figure 1). Increasing SUT expression can lead to increased biomass and yield. For example, the overexpression of SUT1 in *Solanum tuberosum* plants resulted in increased tuber yield and improved growth under both normal and drought conditions [129]. This strategy aims to optimize the source–sink relationship in plants, increasing overall productivity. Sucrose is produced in leaf mesophyll cells (source tissues) and must be transported to various sink tissues, such as roots, fruits, and seeds, where it is used for growth, storage, or metabolic processes. SUTs load sucrose into phloem sieve elements from source cells [130]. This active transport process is required to transport sucrose over long distances through the vascular system. SUTs unload sucrose from the phloem into cells where it can be used or stored. This process is critical for ensuring that developing tissues receive sufficient carbohydrates. Genetic modifications aimed at the overexpression of *SUT* genes have shown significant promise for increasing plant productivity. By increasing the efficiency of sucrose transport, plants can allocate resources better, resulting in improved growth and higher yields. The overexpression of the *SUT1* gene in *S. tuberosum* tubers resulted in increased tuber yields. This finding indicates that a more efficient transport of sucrose may result in a greater distribution of carbohydrates to storage organs. In addition, these transgenic plants showed improved growth under drought conditions. Increased expression of SUT helps maintain an efficient supply of sugars to all parts of the plant, which is especially beneficial under stress conditions when resource allocation is critical [131].

### 4.2. Aquaporins

Aquaporins are membrane proteins that facilitate the transport of water and small solutes across the cell membranes. They play a critical role in maintaining cellular water homeostasis and are required for various physiological processes, including nutrient uptake, cell elongation, and stress responses (Figure 1).

Aquaporins, especially those belonging to the plasma membrane intrinsic protein (PIP) family, have been shown to increase the movement of water into and out of cells, which is critical for maintaining cell turgor and stomatal function. The potential benefits of the overexpression of aquaporin genes in various crops to improve their productivity under water-limited conditions have been demonstrated. For example, transgenic *O. sativa* plants overexpressing the aquaporin gene *OsPIP1;1* showed improved water uptake, photosynthesis, and yield under drought conditions [13,132]. Compared with nontransgenic control plants, these transgenic plants maintained greater leaf water content and photosynthetic activity, resulting in improved growth and productivity. Similarly, studies on other plants have shown that aquaporins may play a significant role in enhancing drought tolerance and photosynthetic efficiency. The overexpression of aquaporin genes in *N. tabacum* and *A. thaliana* resulted in increased water-use efficiency and photosynthetic rates, demonstrating that this strategy is applicable to a variety of plant species [133,134]. By optimizing water transport, aquaporins help maintain the optimal opening and closing of the stomata. This regulation is critical for the efficient uptake and transpiration of CO_2_ and directly affects photosynthesis efficiency.

The understanding of plant aquaporins, especially their role in CO_2_ transport, has been advanced by several key studies. Chen et al. (2023) provided a comprehensive overview of the role of aquaporins in CO_2_ diffusion across biological membranes [135]. Their study highlights the complexity of CO_2_ permeability in membranes and the potential involvement of aquaporins in facilitating CO_2_ transport. They discuss the challenges of measuring CO_2_ permeability and suggest that further research is needed to better understand the molecular mechanisms underlying CO_2_ transport through aquaporins. The authors highlighted the importance of aquaporin expression levels and the role of other membrane proteins and sterols in influencing CO_2_ permeability [135].

Groszmann et al. (2017) investigated the role of aquaporins, focusing on the transport of CO_2_ and water through these channels in plants. They specifically studied the physiological roles of aquaporins and examined how environmental conditions influence their functionality [136]. The effects of altered aquaporin expression on mesophyll conductance and photosynthesis have been investigated. This study used transgenic plants to demonstrate that changes in aquaporin levels correlate with changes in mesophyll conductance and the CO_2_ assimilation rate [136].

Kaldenhoff et al. (2014) [137] reported that aquaporins are involved in CO_2_ transport in plants and suggested that these proteins are required for the efficient transport of CO_2_ across membranes. They discuss the limitations of traditional CO_2_ diffusion models and suggest that aquaporins provide a regulated pathway for CO_2_ transport that is critical for processes such as photosynthesis and respiration. This study highlights the role of certain aquaporins in various plant species and their potential to increase CO_2_ permeability [137].

Uehlein et al. (2003) identified the tobacco aquaporin NtAQP1 as a CO_2_ membrane pore with significant physiological functions [138]. Their study revealed that NtAQP1 facilitates CO_2_ transport by influencing photosynthesis and water-use efficiency in plants. This study is the first to show that aquaporins can function as gas channels, opening new avenues for understanding the role of these proteins in plant physiology.

Several genes encoding aquaporins have been shown to improve the permeability of plants to CO_2_. For example, the genes *AtPIP1;2*, *AtPIP1;3*, and *AtPIP2;6* have been studied in *A. thaliana*, with mixed results regarding their effects on mesophyll conductance and photosynthetic efficiency [139]. Other studies have focused on aquaporins from crops such as *O. sativa* (OsPIP1;2, OsPIP1;3) [13], *Zea mays* (ZmPIP1;5, ZmPIP1;6) [140], *Hordeum vulgare* (HvPIP2;1, HvPIP2;2, HvPIP2;3, HvPIP2;5) [140], and *Setaria italica* (SiPIP2;7) [141]. These studies generally support the role of aquaporins in improving CO_2_ transport, although the results may vary depending on the growth conditions and the specific aquaporin studied.

The oligomeric and phosphorylated states of aquaporins also influence their functionality, which affects their ability to transport CO_2_. For example, phosphorylation can alter the pore size and transport capacity of aquaporins, thereby affecting their effectiveness under different environmental conditions [136].

The overexpression of aquaporins enhances the ability of plants to cope with various abiotic stresses, including salinity, extreme temperature, and drought. Increasing the expression of aquaporins is a promising strategy for improving the photosynthetic performance and water-use efficiency of plants, especially under drought stress [142].

### 4.3. Carbonic Anhydrase

Carbonic anhydrase (CA) promotes the rapid interconversion of CO_2_ and bicarbonate and plays a vital role in maintaining the CO_2_ supply of ribulose-1,5-bisphosphate carboxylase/oxygenase (Rubisco) (Figure 1 and Figure 2). By catalyzing the conversion of these two forms of inorganic carbon, CA provides a steady supply of CO_2_, which is essential for efficient photosynthesis.

CA is critical for increasing the internal CO_2_ concentration in the chloroplast stroma, where Rubisco acts. The increased concentration of CO_2_ around Rubisco helps increase the carboxylation efficiency of the enzyme, thereby increasing the overall rate of photosynthesis. This process is particularly crucial in C_3_ plants, where photorespiration can significantly reduce photosynthetic efficiency by competing with the carboxylation process [143].

Increasing CA activity in transgenic plants has been shown to increase internal CO_2_ concentration, thereby increasing photosynthetic efficiency. For example, transgenic *N. tabacum* plants overexpressing carbonic anhydrase exhibit increased photosynthetic rates and growth under both ambient and elevated CO_2_ conditions [144,145]. These plants demonstrated enhanced adaptation to fluctuating CO_2_ levels, making them more resilient to changing atmospheric conditions.

Increased CA activity can also improve plant tolerance to stress conditions, such as drought and high temperatures, where efficient water use and CO_2_ uptake are critical. By optimizing the internal CO_2_ supply, CA supports sustained photosynthetic activity even under adverse conditions, thus contributing to improved growth and survival rates [146].

CA activity is also associated with improved water-use efficiency. By facilitating the rapid conversion of bicarbonate to CO_2_, CA reduces the need for stomatal opening, thereby minimizing water loss. This function is particularly beneficial under water-limited conditions because it enhances the ability of plants to maintain photosynthetic rates while conserving water.

## 5. Gene Regulation

### 5.1. High Pigment Epistasis 1

The *high pigment epistasis 1 (HPE1*) gene encodes a chloroplast protein that features an RNA recognition motif that is critical for regulating plastid gene expression. Deficiency in *HPE1* affects the expression of nuclear-encoded chlorophyll-related genes, likely through plastid-to-nucleus signaling pathways, which alter chlorophyll metabolism. This deficiency decreases the overall chlorophyll levels but increases the chlorophyll *a*/*b* ratio, leading to a modified composition of the photosynthetic apparatus. A reduction in the total chlorophyll content and the consequent decrease in antenna size result in more efficient light absorption. This adjustment minimizes energy waste and reduces photodamage, thereby enhancing the photosynthetic quantum efficiency of the plant. In *A. thaliana hpe1* mutants, these changes are associated with faster electron transport and increased carbohydrate content, indicating improved photosynthetic performance. Moreover, the improved light absorption and utilization under varying light conditions suggest that the hpe1 mutants can better adapt to and thrive in fluctuating light environments. Ultimately, manipulating the *HPE1* gene could offer strategies for optimizing photosynthetic efficiency and increasing biomass production in crops [147].

### 5.2. Transcription Factors

TFs are proteins that regulate gene expression by binding to specific DNA sequences. They play essential roles in controlling photosynthesis-related gene expression and improving photosynthetic performance (Figure 1).

GLK transcription factors are crucial regulators of chloroplast development and function. Its overexpression in *A. thaliana* and *O. sativa* has been demonstrated to increase chloroplast biogenesis, increase chlorophyll content, and improve photosynthetic efficiency [16]. GLKs activate genes involved in chlorophyll biosynthesis, photosystem assembly, and chloroplast division, thereby increasing photosynthetic capacity. The target genes regulated by Golden2-like (GLK) transcription factors encode proteins essential for photosynthesis, such as those involved in chlorophyll biosynthesis, light absorption, and electron transport. In field-grown *O. sativa*, the expression of *Z. mays GLK* genes increased chlorophyll levels and pigment-protein antenna complexes, increasing light-harvesting efficiency via PSII. Higher levels of xanthophylls in chloroplasts improved energy dissipation as heat achieved, mitigating the effects of photoinhibition under high light conditions. This led to a significant increase in photosynthetic capacity and carbohydrate content and a 30–40% increase in both vegetative biomass and grain productivity [148].

NAC transcription factors are involved in various plant processes, including stress responses and development. Some NAC TFs, such as VND7, have been shown to improve photosynthetic efficiency by regulating the genes involved in chloroplast function and stress tolerance [149]. The overexpression of VND7 in *A. thaliana* enhances photosynthetic performance and biomass production under stress conditions. NAC transcription factors also play crucial roles in secondary cell wall formation, which is essential for plant structural integrity. Additionally, VND7 has been implicated in the differentiation of xylem vessel cells, which are a key component of water and nutrient transport in plants. Studies have indicated that manipulating NAC TFs can lead to improved resilience to biotic and abiotic stresses [150].

AP2/EREBP transcription factors are involved in regulating plant development and stress responses. The overexpression of certain AP2/EREBP TFs, such as dehydration-responsive element-binding protein (DREB), has been shown to improve the photosynthetic efficiency and drought tolerance of transgenic plants [151]. DREBs increase the expression of genes involved in stress responses and photosynthesis, leading to improved plant performance under adverse conditions. These transcription factors also regulate genes involved in cold and heat stress responses, making them versatile tools for improving plant resilience. In addition to affecting photosynthesis, AP2/EREBP TFs influence reproductive development, impacting flowering time and seed set [152].

bZIP transcription factors regulate various physiological processes, including light signaling and stress responses. The overexpression of HY5 (elongated hypocotyl 5), a bZIP transcription factor, in *A. thaliana* has been reported to increase photosynthetic efficiency and biomass production by promoting the expression of light-responsive genes and chlorophyll biosynthesis [153]. bZIP TFs are also critical for orchestrating responses to biotic stresses, such as pathogen attacks, by regulating immune-related genes. HY5 also plays a role in seedling development and photomorphogenesis, influencing how plants adapt to light environments (Figure 1).

### 5.3. MicroRNAs

As mentioned above, miR156, a well-studied miRNA, increases photosynthetic efficiency and biomass in *A. thaliana* and *O. sativa* by regulating SPL genes, which are crucial for leaf development and delayed senescence, thereby promoting chlorophyll accumulation (Figure 1) [154]. Recent research has indicated that the overexpression of miR156 and reduced expression of *SPL13* result in increased levels of DIHYDROFLAVONOL 4-REDUCTASE (DFR), a key enzyme in the phenylpropanoid pathway responsible for the production of anthocyanin precursors [155]. miR396-mediated regulation of GRF genes enhances leaf architecture and chloroplast development, thereby improving photosynthetic performance. Chloroplasts are cellular organelles where photosynthesis occurs, and their development is crucial for efficient light capture and conversion into chemical energy. The enhanced leaf architecture, influenced by miR396, provides an optimal arrangement for light interception, further increasing the photosynthetic efficiency [156]. miR319 targets the transcription factor TEOSINTE BRANCHED/CYCLOIDEA/PCF (TCP), which plays a role in the regulation of leaf development. In *N. tabacum*, overexpression of miR319 results in increased leaf size, increased chlorophyll content, and increased photosynthesis. TCP transcription factors are involved in the modulation of cell proliferation in leaves, leading to an increase in leaf area and, thus, a larger photosynthetic surface area. Additionally, a relatively high chlorophyll content improves the efficiency of light absorption and conversion, which are critical for photosynthesis [157,158].

miR408 is known for its role in increasing photosynthetic activity and improving plant resistance to environmental stresses, such as drought and high salinity. The overexpression of miR408 increases photosynthetic efficiency by increasing the expression of plastocyanin, a protein involved in the electron transport chain of photosynthesis. This increase enhances the efficiency of electron transport, thus improving the overall photosynthetic output. Furthermore, miR408 enhances the ability to cope with environmental stress by modulating stress-responsive genes, thereby maintaining photosynthetic activity under adverse conditions [159,160].

miR162 targets DICER-LIKE 1 (DCL1), a key enzyme in the biogenesis of other miRNAs. Modulating miR162 levels can influence the overall miRNA regulatory network, impacting various processes, including photosynthesis and stress responses. In *O. sativa*, the overexpression of miR162 increases the photosynthetic efficiency and yield under stress conditions [161,162].

miR397 targets laccase genes involved in lignin biosynthesis and cell wall modification. The overexpression of miR397 leads to decreased lignin content, which facilitates improved photosynthetic efficiency and biomass production. Lignin is a complex polymer in the plant cell wall that provides structural support but can also restrict cell wall flexibility and water transport. A reduced lignin content resulting from miR397 overexpression enhances the efficiency of water and nutrient transport within the plant, thereby improving photosynthetic efficiency. In *O. sativa*, miR397 overexpression enhances grain yield and biomass by promoting photosynthesis and reducing lignin content [163,164]. miR172 targets APETALA2 (AP2)-like transcription factors that are involved in the regulation of flowering time and development. The overexpression of miR172 can influence leaf morphology and enhance photosynthetic efficiency by modulating the developmental timing and architectural traits of plants. This miRNA affects various developmental processes that contribute to improved growth and productivity, making it a valuable target for optimizing photosynthetic efficiency and plant performance [165,166].

miRNAs, such as miR156, miR396, miR319, miR408, miR162, miR397, and miR172, play important roles in regulating photosynthesis and plant productivity. By targeting specific genes involved in leaf development, chlorophyll accumulation, cell proliferation, and stress responses, these miRNAs can significantly increase photosynthetic efficiency and biomass production.

## 6. Environmental Stressors Limiting Photosynthesis

### 6.1. Temperature Stress

Heat stress significantly damages the photosynthetic apparatus [167]. High temperatures disrupt the thylakoid membrane, affecting electron transporters and enzymes, particularly PSI, PSII, Cyt *b6f*, and Rubisco [168,169,170]. Increased thylakoid membrane fluidity under heat stress compromises the light-harvesting complex and the structural integrity of PSII [171]. Heat stress also inhibits chlorophyll biosynthesis by reducing the activity of biosynthetic enzymes [172,173]. For example, high temperatures can impair the activity of 5-aminolevulinate dehydratase, which is involved in the pyrrole biosynthetic pathway, or suppress protochlorophyllide biosynthesis [174].

Cold stress reduces thylakoid membrane fluidity, affecting electron transport and energy transfer efficiency. It also inhibits Calvin cycle enzymes, further limiting photosynthesis [175]. Cold stress represses the expression of genes associated with the photosynthetic apparatus. Furthermore, photoinhibition at low temperatures results in light-induced damage to PSII, exceeding its repair capacity and reducing photosynthetic efficiency [176].

Photosynthesis is inhibited by low temperatures, which can cause the formation of ROS and lead to oxidative stress. This stress damages the photosynthetic apparatus, particularly PSII. Low temperatures can lead to a decrease in the fluidity of the thylakoid membrane, affecting electron transport and reducing the efficiency of energy transfer. Cold stress also inhibits the activity of enzymes involved in the Calvin cycle, such as Rubisco, further limiting photosynthetic performance.

At low temperatures, the expression of genes related to the photosynthetic machinery is often downregulated, which compromises the ability to carry out photosynthesis effectively. Cold stress can also lead to photoinhibition, where light-induced damage to PSII exceeds its repair capacity, exacerbating the decline in photosynthetic efficiency [177].

To increase resistance to low-temperature stress, various genes that can increase cold tolerance in plants have been identified. For example, overexpression of *CBF (C-repeat binding factor)* genes has been shown to improve cold tolerance by regulating the expression of *cold-responsive (COR)* genes [178]. These COR genes help stabilize the cellular structure and function under cold-stress conditions.

Additionally, the overexpression of genes encoding antioxidant enzymes, such as SOD and CAT, can help mitigate oxidative stress caused by low temperatures by scavenging ROS [179]. Another promising approach is the manipulation of genes involved in the unsaturation of membrane lipids, which can increase membrane fluidity and maintain photosynthetic efficiency at low temperatures [180].

### 6.2. Light Stress

Both high and low light intensity stress photosynthesis. High light intensity causes stress due to excessive light absorption, which reduces photosynthesis and quantum performance, even after the plants return to normal light. This excess energy generates ROS, damaging PSII, proteins, lipids, and nucleic acids. UV radiation exacerbates this damage by directly affecting the DNA and proteins in chloroplasts, further impairing photosynthetic efficiency and plant growth [181].

Low light reduces intercellular CO_2_ in plant leaves, decreasing photosynthesis. It also restricts other photosynthetic parameters, including transpiration rate, stomatal conductance, net photosynthetic rate, maximum PSII quantum efficiency, and water-use efficiency [182].

To increase resistance to light stress and UV radiation, certain genes can be targeted. The overexpression of genes encoding ROS-scavenging enzymes, such as SOD, CAT, and ascorbate peroxidase (APX), enhances tolerance to high light and UV stress by mitigating oxidative damage [183,184]. Upregulating genes involved in the synthesis of photoprotective pigments, such as carotenoids and flavonoids, protects the photosynthetic machinery from light-induced damage [184,185,186,187].

Additionally, overexpression of genes related to PSII repair mechanisms, such as PsbS, which is involved in nonphotochemical quenching (NPQ), enhances the recovery and stability of the photosynthetic apparatus under stress conditions [41].

### 6.3. Water and Salt Stresses

Water deficiency significantly damages the thylakoid membrane and decreases chlorophyll content in plants, thereby limiting the efficiency of the photosynthetic apparatus [176]. For example, water deficiency, by reducing its quantum yield, negatively affects PSII [188]. Additionally, it leads to the destruction of the thylakoid membrane and chlorophyll and, as a result, reduces the chlorophyll content of the plant [176].

Many studies have shown that the main and initial response of plants to drought stress is the closing of stomata, which leads to a net decrease in CO_2_ absorption, ultimately leading to a decrease in photosynthesis. Under drought stress, in addition to stomatal closure, other factors, called nonstomatal mechanisms, also reduce the rate of photosynthesis. For example, reduced CO_2_ uptake by stomatal closure modified the activities of carbon-fixing enzymes, resulting in membrane disruption and decreased ATP synthesis, which eventually limited the function of Rubisco by altering the regeneration of RuBP [189]. Additionally, under water deficit conditions, NADP^+^ regeneration is affected, which causes electron leakage and an excessive decline in the electron transport chain [190].

Salinity causes osmotic stress in plants, which reduces photosynthesis through its ionic impact on the structure of intracellular organelles and inhibition of metabolic processes [191]. Under salt stress, damage to the thylakoid membrane is caused by an increase in the levels of ions such as Na^+^ and Cl^−^ in the chloroplast [192]. Additionally, high levels of mineral salts in the thylakoid membrane deactivate phosphorylation and inhibit electron transfer [193]. The photosynthetic reduction caused by salt is closely related to several factors, including inhibition of the electron transfer chain from PSII to PSI, changes in enzyme activities, inhibition of chlorophyll biosynthesis, damage to the photosynthetic apparatus, nonphotochemical loss of thermal energy, changes in gene expression, and a reduction in the supply of CO_2_ due to stomatal closure [194].

To mitigate these adverse effects and enhance resistance, several genes have been identified and used in genetic engineering. Genes *such as dehydration-responsive element-binding protein 1A* (*DREB1A*) [195], *sodium/hydrogen antiporter* (*NHX1*) [196], *pyrroline-5-carboxylate synthetase (P5CS)* [197], and *high-affinity potassium transporter (HKT1*) are involved in stress responses and have shown potential for conferring drought and salinity tolerance [198]. The overexpression of these genes can increase osmotic adjustment, ion homeostasis, and ROS scavenging, thereby improving photosynthetic efficiency and plant resilience under osmotic conditions [199].

### 6.4. Biotic Stress

Photosynthesis is highly susceptible to biotic stresses from viruses, bacteria, fungi, and insects and can severely impair photosynthetic efficiency and overall plant health [200]. Viruses such as tobacco mosaic virus (TMV) and tomato yellow leaf curl virus (TYLCV) are known to disrupt the photosynthetic machinery in plants. TMV causes chlorosis and mosaic patterns in leaves by interfering with chloroplast structure and function. This virus disrupts the electron transport chain within chloroplasts, which diminishes the ability to convert light energy into chemical energy. Similarly, TYLCV affects the transcription and translation of chloroplast-encoded proteins, further impairing the photosynthetic apparatus [201].

To combat these viral infections, certain genes that confer resistance have been identified. The *N* gene from *Nicotiana glutinosa* encodes a TIR-NBS-LRR class resistance protein that recognizes the TMV replicase protein, triggering a hypersensitive response to the virus [202]. Additionally, the *Ty-1* and *Ty-3* genes confer resistance to TYLCV by encoding proteins involved in RNA silencing pathways, which degrade viral RNA and inhibit virus replication [203].

Bacterial pathogens, such as *Pseudomonas syringae,* also pose a significant threat to photosynthesis. These bacteria secrete effector proteins into host cells that target chloroplasts, leading to a reduction in photosynthetic efficiency and increased susceptibility to oxidative stress. For example, the effector protein HopI1 from *P. syringae* modifies the chloroplast structure, disrupting its function [204]. To counter bacterial infections, genes such as *RPM1* and *RPS2* play critical roles in *A. thaliana*. *RPM1* encodes a nucleotide-binding site-leucine-rich repeat (NBS-LRR) protein that detects bacterial effectors and activates immune responses. Similarly, RPS2 recognizes specific bacterial effectors and initiates defense mechanisms [205].

Fungal pathogens, including *Botrytis cinerea* and *Blumeria graminis*, infect plant leaves and produce toxins that degrade chlorophyll and disrupt the photosynthetic electron transport chain [206]. These fungi also induce oxidative damage through the production of ROS. Genetic resistance to fungal infections can be enhanced by genes such as *RGA2* in *O. saitva* and *RPW8* in *A. thaliana*. *RGA2* encodes a protein that recognizes pathogen-associated molecular patterns (PAMPs) from the *O. saitva* blast fungus *Magnaporthe oryzae*, triggering immune responses. RPW8 provides broad-spectrum resistance to powdery mildew fungi by activating localized cell death around infection sites, preventing the spread of the pathogen [207].

Insect herbivory is another significant biotic stress affecting photosynthesis. Insects, such as aphids and whiteflies, feed on plant sap, causing physical damage to leaves and the transmission of viral and bacterial pathogens. This feeding activity leads to chlorosis, reduced chlorophyll content, and impaired photosynthetic capacity [208]. Genes such as *Mi-1.2* from *S. lycopersicum* and *Vat* from *Cucumis melo* have been shown to confer resistance against insect pests. *Mi-1.2* encodes a protein that activates plant defenses in response to insect feeding, whereas *Vat* interferes with aphid-feeding behavior and reduces the transmission of viruses [209].

### 6.5. Cross-Adaptation

Achieving resistance to multiple types of stress in plants involves the manipulation of several key genes associated with various physiological pathways. These pathways include those involved in abiotic stress (e.g., drought, salinity, temperature extremes, and heavy metal toxicity) and biotic stress (e.g., pathogens and pests). The concept of cross-adaptation implies that by transforming certain genes, plants can increase their resilience to a range of stresses simultaneously. Adaptation to factors of different natures is provided by three large groups of genes: genes encoding molecular chaperones, genes encoding chemical chaperones or compatible osmolytes, and genes encoding enzymatic and nonenzymatic antioxidants. Induction of the expression of the listed genes prevents damage to photosynthetic function and maintains plant productivity under stress conditions [210].

Heat shock proteins (HSPs), such as HSP70 and HSP90, function as molecular chaperones, assisting in protein folding and protection under stress. The overexpression of HSP70 in plants such as *A. thaliana* enhances tolerance to heat, drought, and salinity [211].

Genes involved in the biosynthesis of chemical chaperones or compatible osmolytes (sugars, sugar alcohols, amino acids, and quaternary amines) are also essential for stress tolerance. For example, genes such as *P5CS* and *betaine aldehyde dehydrogenase (BADH)* help stabilize proteins and cell structures during stress [212]. P5CS, which is involved in proline biosynthesis, accumulates under drought and salinity stress conditions, increasing tolerance when overexpressed in plants such as *N. tabacum* and *O. sativa* [213,214]. BADH, which is involved in glycine betaine biosynthesis, aids in osmotic adjustment under stress conditions, with transgenic rice overexpressing BADH showing improved tolerance to salt and drought stress [65].

Antioxidant enzymes mitigate damage from ROS generated during stress. The overexpression of SOD in plants such as *S. lycopersicum* and *N. tabacum* results in the conversion of superoxide radicals to less harmful hydrogen peroxide, increasing tolerance to multiple stresses, including drought, salinity, and heavy metals [215]. Similarly, APX catalyzes the conversion of hydrogen peroxide to water, thereby reducing oxidative damage. Transgenic plants overexpressing *APX* exhibit improved resistance to abiotic stresses, such as drought, salinity, and oxidative stress [216,217].

Under conditions of water deficiency, salt stress, and heavy metal pollution, maintaining cellular ionic homeostasis is especially important. Under conditions of water deficiency, salt stress, and heavy metal pollution, maintaining cellular ionic homeostasis is extremely important. Ion transporters and ion channels play key roles in this process. The NHX family, which is involved in sodium and potassium homeostasis, improves tolerance to salt stress and drought when *NHX1* is overexpressed in *A. thaliana* [218]. The HKT family regulates sodium and potassium balance, with high-affinity K+ transporter (*HKT1)* overexpression in *O. sativa* enhancing salt tolerance and reducing sodium accumulation in plant tissues [219]

TF genes, which regulate the expression of not only one gene but also an entire cassette of genes, play a special role in cross-adaptation. Notably, the DREB/CBF family of TFs plays a significant role in abiotic stress responses, such as those to drought, cold, and salinity. The overexpression of *DREB1A* enhances tolerance to drought, high salinity, and cold [220]. Similarly, NAC family TFs, such as ANAC019, ANAC055, and ANAC072, respond to drought, salinity, and pathogen attacks. The overexpression of these genes in *A. thaliana* increases tolerance to drought and salt stress, as well as resistance to certain pathogens [221].

## 7. Conclusions

Advances in plant genetic engineering and genome editing have provided promising strategies to increase photosynthetic activity under different environmental conditions. Among the biological targets for genetic manipulation, considerable attention has been paid to increasing the activity of RBC, an essential enzyme involved in CO_2_ fixation. Despite its central role, the catalytic efficiency of Rubisco is relatively low, which is a bottleneck in the efficiency of photosynthesis. Several strategies have been proposed to address this issue, including increasing the expression of large and small Rubisco subunits and engineering mechanisms to increase CO_2_ concentrations around its active sites. Enhancing photosynthesis in C_3_ plants by using genes from C_4_ plants is another promising approach. This method aims to improve the use of water and carbon dioxide, thereby increasing plant productivity. Another critical aspect of enhancing plant photosynthesis involves the development of artificial photorespiration pathways that can process energy more efficiently and reduce the production of toxic byproducts, such as 2-phosphoglycolate.

In addition to improving photosynthesis-related enzymes, increasing plant resistance to various stressors is crucial. Improvements in the expression of chaperone proteins such as HSPs, TFs, and associated microRNAs, as well as osmoprotectants and antioxidant enzymes, can significantly improve plant stress tolerance. These modifications may improve cross-adaptation to concurrent stressors in natural settings.

Overall, genetic engineering and biotechnology offer enormous potential for improving photosynthetic performance and adapting plants to changing environmental conditions. These advances have important implications for ensuring food security and promoting sustainable agriculture in the face of global climate change.

## Figures and Tables

**Figure 1 cells-13-01319-f001:**
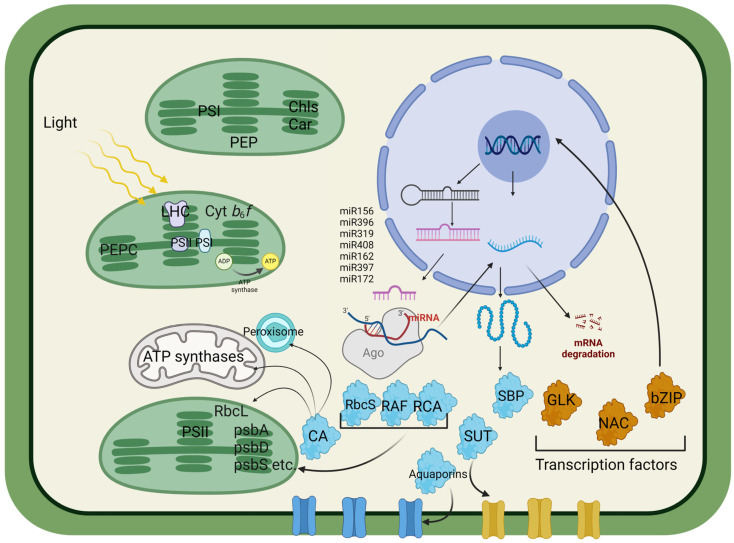
Approximate diagram of the possible regulation of photosynthetic activity and its relationship with nuclear-encoded proteins, indicating the most promising targets for plant transformation to increase photosynthetic processes and productivity. Phosphoenolpyruvate carboxylase (PEPC) catalyzes the conversion of phosphoenolpyruvate (PEP) and bicarbonate (HCO_3_^−^) to oxaloacetate during C_4_ and CAM photosynthesis, thereby increasing CO_2_ fixation. The cytochrome *b_6_f* complex (Cyt *b_6_f*) plays a crucial role in the electron transport chain between PSII and PSI, contributing to the generation of the proton gradient used for ATP synthesis. ATP synthase utilizes a proton gradient to synthesize ATP from ADP and inorganic phosphate. Chlorophylls (Chls) and carotenoids (Cars) are pigments involved in light absorption and protection against photodamage. Carbonic anhydrase (CA) catalyzes the interconversion of CO_2_ and HCO_3_^−^, facilitating efficient CO_2_ utilization for the Calvin cycle. The Rubisco large subunit (RbcL) and small subunit (RbcS) are components of Rubisco, the enzyme responsible for CO_2_ fixation in the Calvin cycle. The Rubisco assembly factor (RF) assists in the assembly and activation of Rubisco, thereby increasing its efficiency. Rubisco activase (RCA) facilitates Rubisco activation by removing inhibitory sugar-phosphate compounds from its active sites. Transcription factors (TFs), such as Golden2-like (GLK), regulate chloroplast development and photosynthesis-related gene expression. NAC (NAM, ATAF, and CUC) factors are involved in various aspects of plant development and stress responses. Basic leucine zippers (bZIPs) play roles in the stress response, hormone signaling, and development. SQUAMOSA promoter binding protein-like (SBP) is involved in flower development and phase transition. The sucrose transporter (SUT) facilitates the transport of sucrose across cellular membranes. Aquaporins facilitate water transport across cell membranes, impacting turgor pressure and cell expansion. MicroRNAs (miRNAs) are small noncoding RNA molecules that regulate gene expression posttranscriptionally by binding to complementary sequences on target mRNAs. Argonaute (Ago), a part of the RNA-induced silencing complex (RISC), guides miRNA to its target mRNA, leading to mRNA degradation or translational repression. miRNAs, such as miR156, miR396, miR319, miR408, miR162, miR397, and miR172, regulate various aspects of plant growth and development by targeting specific mRNAs. Peroxisomes, organelles involved in the oxidative metabolism and detoxification of ROS, play a role in cellular signaling and stress responses. Chloroplast–nucleus signaling is characterized by feedback mechanisms in which changes in chloroplast function affect nuclear gene expression, influencing overall plant responses to environmental factors.

**Figure 2 cells-13-01319-f002:**
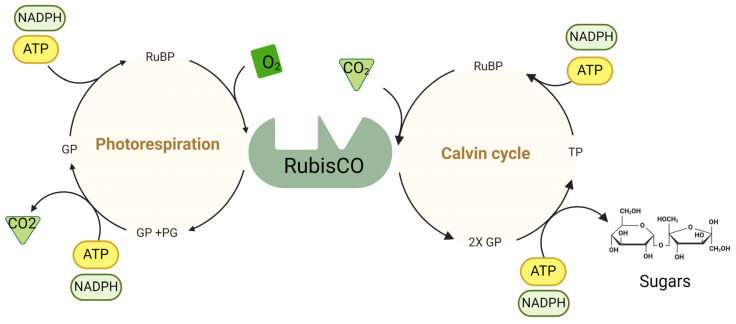
Calvin cycle: Rubisco catalyzes the carboxylation of ribulose-1,5-bisphosphate (RuBP) to fix CO_2_, forming 3-phosphoglycerate (GP), the first stable product. GP is then converted to triose phosphate (TP) via ATP and NADPH. TP can regenerate RuBP or form sugars such as glucose and fructose. ATP provides energy, and NADPH supplies reducing power for these conversions. Photorespiration: When Rubisco reacts with O_2_ instead of CO_2_, RuBP is oxygenated, producing GP and phosphoglycolate (PG). PG is processed via the photorespiratory pathway to recover carbon and return it to the Calvin cycle. ATP and NADPH are utilized to convert the intermediates back into GP. O_2_ competes with CO_2_ at the Rubisco active site, leading to photorespiration. ATP and NADPH generated during the light-dependent reactions of photosynthesis support both the Calvin cycle and photorespiration.

## Data Availability

The data that support the findings of this study are available upon request from the corresponding author.

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
