# Peer review of "Enhancing Photosynthesis and Plant Productivity through Genetic Modification"

_cells, 2024, doi:10.3390/cells13161319_

Round 1
Reviewer 1 Report
Comments and Suggestions for Authors
The Review article by Nazari summarizes various prospects for improving photosynthesis via genetic manipulation (e.g., light harvest, electron transport, Rubisco, other Calvin cycle enzymes, transcription factor, MicroRNA and so on), and highlights key points for future research and development. Currently, there is a great focus on strategies for improving photosynthesis to increase food and fuel productions. Thus, the present topic would be attractive to many readers. Although most of sections are correctly described, the content of the strategy to enhance photosynthetic capacity is biased and thus needs significant revision. In addition, the structure of the manuscript needs significant revision.
(1) In the Introduction section, studies targeting enhanced CO2 supply should also be introduced. In other words, strategies to enhance CO2 transport capacity by 1) promoting stomatal conductance and 2) enhancing mesophyll conductance via aquaporin and carbonic anhydrase should also be mentioned. Specifically, after L76, 1) enhancement of photosynthetic capacity by promoting stomatal conductance should be described, followed by 2) enhancement of photosynthetic capacity by enhancing mesophyll conductance (aquaporin and carbonic anhydrase).
For example,
For 1) promoting stomatal conductance, it would be good to describe the following papers: Papanatsiou et al. (2019) Science, 363(6434), 1456-1459; Yamori et al. (2020) Plant, Cell & Environment 43, 1230-1240; Kimura et al. (2020) Journal of Experimental Botany 71, 2339-2350; Shimadzu et al. (2019) Frontiers in Plant Science, 10, 1512; Sakoda et al. (2020) Frontiers in Plant Science, 11, 1609.
For 2) enhancing mesophyll conductance via aquaporin, aquaporin could contribute to the enhancement of photosynthetic capacity, since it allows not only H2O but also CO2 to pass through. Thus, it would be good to describe it well with citations of the following papers: Xu et al. (2019). Journal of Experimental Botany, 70(2), 671-681; Tsuchihira et al, (2010). Tree physiology, 30(3), 417-430; Hanba et al. (2004) Plant and Cell Physiology, 45(5), 521-529.
(2). The categorization of “2.1. chloroplast genes” and “2.2. nuclear genes” is wrong, and thus needs to be corrected. In this paper, the classification for “2.1. chloroplast genes” is not “Chloroplast-encoded-genes”, but it seems to be “Genes functioned in Chloroplast”. However, “Genes functioned in Chloroplast” are also included in “2.2. nuclear genes”. Taken together, the overall structure needs to be substantially revised.
(3) A section of 1. Cyclic ETR and 2. CO2 supply via Stomata should be added before 2.1.7. Photorespiratory.
In particular, the following paper should be cited to describe the Physiological Functions of Cyclic Electron Transport Around Photosystem I: Yamori & Shikanai T (2016) Annual Review of Plant Biology 67:81-106. In addition, the following two examples of enhanced PSI activity should be cited: Basso et al. (2022) Plant Physiology, 189, 375-387; Wada et al. (2018) Plant physiology. 176, 1509–1518.
(4) L550-609: As Rubisco activase should not be a section of “2.2.1 Rubisco assembly factor”, the section/section title should be edited. Moreover, the recent review for Rubisco activase should be described and cited here: Qu et al. (2023) Journal of Experimental Botany. 74: 591-599.
(5) A section for Aquaporin should added before/after the “2.2.5. Carbonic Anhydrase (CA)”.
Aquaporins are also knows as CO2 transporters. It would be good to describe its information here. Ther recent review is, for example, Chen et al. (2023) Frontiers in Physiology, 14, 1205290. The following papers are well cited: Groszmann et al. (2017) Plant, Cell & Environment, 40(6), 938-961; Kaldenhoff et al. (2014)Biochimica et Biophysica Acta (BBA)-General Subjects, 1840(5), 1592-1595; Uehlein et al. (2003) Nature, 425(6959), 734-737.
(6) The section for “3. High-Temperature, Light, Drought, and Salinity Stress Limiting Photosynthesis“ is not well summarized in the current manuscript. As the content of this section is not enough, I think that it would be good to delete this section. Instead, it might be good to point it out briefly in “4. Conclusion”.
For example,,,, Plants in natural environments must cope with diverse, highly dynamic, and unpredictable conditions (Yamori, 2016, Journal of Plant Research 129:379–395). The effects of global warming on plant growth and yield have attracted considerable attention (for a review, see Yamori et al. (2014) Photosynthesis Research 119, 101-117). [Then, you can briefly insist on the consideration for strategies for enhancing photosynthesis.]
Minor points
1) L46-47: The description is wrong, since Light dependent reactions via thioredoxins are included in the Calvin cycle. Please edit this sentence.
2) L71, L298-299: please cite the appropriate paper (Yamori et al. 2016 Plant, Cell & Environment 39, 80-87.)
Comments on the Quality of English LanguageMinor editing of English language required
Author Response
The authors sincerely thank the distinguished reviewers for reviewing our data, providing useful recommendations, and providing friendly criticism.
Reviewer 1
1) In the Introduction section, studies targeting enhanced CO2 supply should also be introduced. In other words, strategies to enhance CO2 transport capacity by 1) promoting stomatal conductance and 2) enhancing mesophyll conductance via aquaporin and carbonic anhydrase should also be mentioned. Specifically, after L76, 1) enhancement of photosynthetic capacity by promoting stomatal conductance should be described, followed by 2) enhancement of photosynthetic capacity by enhancing mesophyll conductance (aquaporin and carbonic anhydrase).
For example,
For 1) promoting stomatal conductance, it would be good to describe the following papers: Papanatsiou et al. (2019) Science, 363(6434), 1456-1459;
Yamori et al. (2020) Plant, Cell & Environment 43, 1230-1240;
Kimura et al. (2020) Journal of Experimental Botany 71, 2339-2350;
Shimadzu et al. (2019) Frontiers in Plant Science, 10, 1512;
Sakoda et al. (2020) Frontiers in Plant Science, 11, 1609.
For 2) enhancing mesophyll conductance via aquaporin, aquaporin could contribute to the enhancement of photosynthetic capacity since it allows not only H2O but also CO2 to pass through. Thus, it would be good to describe it well with citations of the following papers:
Xu et al. (2019). Journal of Experimental Botany, 70(2), 671-681;
Tsuchihira et al, (2010). Tree physiology, 30(3), 417-430;
Hanba et al. (2004) Plant and Cell Physiology, 45(5), 521-529.
Answer: We agree. Some new paragraphs have been added to the text.
Stomatal conductance plays a critical role in photosynthesis by regulating the exchange of gases, primarily carbon dioxide (CO2) and water vapour, between the plant and its environment. Stomatal conductance is influenced by the density and opening size of stomata. One approach to increase stomatal conductance is through the genetic manipulation of key regulatory genes involved in stomatal development and functioning, such as EPF (epidermal patterning factor) Sakoda, K., Yamori, W., Shimada, T., Sugano, S. S., Hara-Nishimura, I., & Tanaka, Y. (2020). Higher stomatal density improves photosynthetic induction and biomass production in Arabidopsis under fluctuating light. Frontiers in Plant Science, 11, 589603 and SLAC1 (Slow Anion Channel-Associated 1) Yamori, W., Kusumi, K., Iba, K., & Terashima, I. (2020). Increased stomatal conductance induces rapid changes in the photosynthetic rate in response to naturally fluctuating light conditions in rice. Plant, Cell & Environment, 43(5), 1230-1240.) For instance, the overexpression of EPF can lead to an increased number of stomata, while the modification of SLAC1 can result in stomata that remain open for longer periods, thereby allowing more CO2 to enter the leaf (Kimura, H., Hashimoto-Sugimoto, M., Iba, K., Terashima, I., & Yamori, W. (2020). Improved stomatal opening enhances the photosynthetic rate and biomass production under fluctuating light. Journal of experimental botany, 71(7), 2339-2350. Sakoda, K., Yamori, W., Shimada, T., Sugano, S. S., Hara-Nishimura, I., & Tanaka, Y. (2020). Higher stomatal density improves photosynthetic induction and biomass production in Arabidopsis under fluctuating light. Frontiers in Plant Science, 11, 589603..
Mesophyll conductance is another critical factor influencing photosynthetic efficiency and involves the movement of CO2 from the intercellular air spaces within the leaf to the chloroplasts where photosynthesis occurs. Enhancing mesophyll conductance can significantly boost photosynthetic capacity, and genetic modification offers a powerful tool to achieve this goal. Two key components in this process are aquaporins and carbonic anhydrase, both of which facilitate the diffusion of CO2 through the mesophyll Tsuchihira, A., Hanba, Y. T., Kato, N., Doi, T., Kawazu, T., & Maeshima, M. (2010). Effect of radish plasma membrane aquaporin overexpression on the water-use efficiency, photosynthesis and growth of Eucalyptus trees. Tree physiology, 30(3), 417-430.. Hanba, Y. T., Shibasaka, M., Hayashi, Y., Hayakawa, T., Kasamo, K., Terashima, I., & Katsuhara, M. (2004). Overexpression of the barley aquaporin HvPIP2; 1 increases internal CO2 conductance and CO2 assimilation in the leaves of transgenic rice plants. Plant and Cell Physiology, 45(5), 521-529.
Aquaporins are membrane proteins that form channels facilitating the movement of water and small solutes, including CO2, across cell membranes. By increasing the expression of specific aquaporins, researchers can increase the diffusion rate of CO2 within the leaf mesophyll, thus accelerating its delivery to chloroplasts. Studies have shown that plants with higher aquaporin activity exhibit increased photosynthetic rates, particularly under conditions where internal CO2 diffusion is a limiting factor Xu, F., Wang, K., Yuan, W., Xu, W., Liu, S., Kronzucker, H. J., ... & Zhu, Y. (2019). Overexpression of rice aquaporin OsPIP1; 2 improves yield by enhancing mesophyll CO2 conductance and phloem sucrose transport. Journal of Experimental Botany, 70(2), 671-681.. Hanba, Y. T., Shibasaka, M., Hayashi, Y., Hayakawa, T., Kasamo, K., Terashima, I., & Katsuhara, M. (2004). Overexpression of the barley aquaporin HvPIP2; 1 increases internal CO2 conductance and CO2 assimilation in the leaves of transgenic rice plants. Plant and Cell Physiology, 45(5), 521-529. Increasing the expression of aquaporins can improve water use efficiency and photosynthetic performance, especially under stress conditions such as drought. Transgenic Oryza sativa plants overexpressing the aquaporin gene OsPIP1;1 demonstrated improved water uptake, photosynthesis, and yield under water-limited conditions [8]. This strategy aims to enhance the plant's ability to maintain photosynthetic efficiency during periods of water stress.
Carbonic anhydrase is an enzyme that catalyzes the rapid interconversion of CO2 and bicarbonate. By enhancing the expression of carbonic anhydrase, the availability of CO2 in mesophyll cells can increase, further promoting efficient photosynthesis. Genetic modifications that upregulate carbonic anhydrase can lead to a more rapid conversion of bicarbonate to CO2, ensuring a steady supply of CO2 for the photosynthetic machinery Rudenko, N. N., Ignatova, L. K., Fedorchuk, T. P., & Ivanov, B. N. (2015). Carbonic anhydrases in photosynthetic cells of higher plants. Biochemistry (Moscow), 80, 674-687. Momayyezi, M., McKown, A. D., Bell, S. C., & Guy, R. D. (2020). Emerging roles for carbonic anhydrase in mesophyll conductance and photosynthesis. The Plant Journal, 101(4), 831-844.. This approach is particularly advantageous in environments where CO2 availability within the leaf is constrained, such as during high rates of photosynthesis or in densely packed foliage Momayyezi, M., McKown, A. D., Bell, S. C., & Guy, R. D. (2020). Emerging roles for carbonic anhydrase in mesophyll conductance and photosynthesis. The Plant Journal, 101(4), 831-844..
2). The categorization of “2.1. chloroplast genes” and “2.2. nuclear genes” is wrong, and thus needs to be corrected. In this paper, the classification for “2.1. Chloroplast genes” are not “chloroplast-encoded genes”, but they seem to be “genes functioned in the chloroplast”. However, “Genes functioning in the chloroplast” was also included in “2.2. nuclear genes”. Taken together, the overall structure needs to be substantially revised.
Answer: We improved the section order.
- Introduction
- Chloroplast component modifications
2.1. Optimization of Light Harvesting and Pigment Content
2.2. Photosystem II
2.3. Cytochrome b6f Complex
2.4. Photosystem I
2.5. Electron Transport Chain
2.6. Chloroplast ATP synthase
- Carbon Assimilation Efficiency
3.1. RuBisCO as a Target to Improve Carbon Assimilation Efficiency
3.2. Rubisco Assembly Factors
3.3. Sedoheptulose-1,7-bisphosphatase (SBPase)
3.4. Gene C4 type of photosynthesis
3.5. Photorespiration
- Cellular transport and regulation
4.1. Sucrose Transporters
4.2. Aquaporins
4.3. Carbonic Anhydrase (CA)
- Gene Regulation
5.1. High-Pigment Epistasis 1 (HPE1)
5.2. Transcription Factors (TFs)
5.3. MicroRNAs
- Environmental Stressors Limiting Photosynthesis
6.1. Temperature stress
6.2. Light stress
6.3. Water and Salt Stresses
6.4. Biotic stress
6.5. Cross-adaptation
- Conclusions
3) A section of 1. Cyclic ETR and 2. CO2 via Stomata should be added before 2.1.7. Photorespiratory.
Answer: This has been done.
In particular, the following paper should be cited to describe the Physiological Functions of Cyclic Electron Transport Around Photosystems I:
Yamori W, Shikanai T. Physiological functions of cyclic electron transport around photosystem I in sustaining photosynthesis and plant growth. Annual review of plant biology. 2016 Apr 29;67(1):81-106.
In addition, the following two examples of enhanced PSI activity should be cited:
Basso, L., Sakoda, K., Kobayashi, R., Yamori, W. and Shikanai, T., 2022. Flavodiiron proteins enhance the rate of CO2 assimilation in Arabidopsis under fluctuating light intensity. Plant Physiology, 189(1), pp.375-387.
Wada, S., Yamamoto, H., Suzuki, Y., Yamori, W., Shikanai, T. and Makino, A., 2018. Flavodiiron protein substitutes for cyclic electron flow without competing CO2 assimilation in rice. Plant Physiology, 176(2), pp.1509-1518.
Answer: We agree, and paragraphs to the Electon transport section were added to the MS: “The study by Yamori and Shikanai (2016) provides a comprehensive review of the critical roles played by cyclic electron transport (CET) around photosystem I (PSI) in plants. This process is pivotal for the stabilization and optimization of photosynthesis, particularly under fluctuating environmental conditions. CET contributes to the generation of a proton gradient across the thylakoid membrane, which is essential for ATP synthesis, thereby supporting the Calvin cycle and other energy-demanding processes within the chloroplast. The authors highlight the adaptive advantages of CET, allowing plants to manage the balance between ATP and NADPH production and mitigating the formation of reactive oxygen species (ROS), which can cause cellular damage. This review underscores the integral role of CET in enhancing plant growth and productivity, especially under stress conditions such as drought or high light intensity.
Building on the foundational understanding provided by Yamori and Shikanai, recent studies have further elucidated the mechanisms and benefits of enhanced PSI activity in plants. For instance, Basso et al. (2022) investigated the role of flavodiiron proteins (FDPs) in Arabidopsis thaliana under fluctuating light conditions. Their study, titled "Flavodiiron proteins enhance the rate of CO2 assimilation in Arabidopsis under fluctuating light intensity," demonstrated that FDPs can significantly boost the rate of CO2 assimilation. This enhancement occurs by facilitating alternative electron flow, which helps to quickly dissipate excess energy and reduce the buildup of ROS during sudden changes in light intensity. As a result, plants with elevated FDP levels show improved photosynthetic efficiency and better growth performance in environments with variable light conditions, reflecting the practical applications of modulating PSI activity for agricultural productivity. The key genes studied in this work included FLV1 and FLV3, which encode the flavodiiron proteins critical for this process.
Similarly, the research by Wada et al. (2018) provides additional insights into the functionality of FDPs in rice, another crucial crop species. Their paper, "Flavodiiron protein substitutes for cyclic electron flow without competing CO2 assimilation in rice," explores how FDPs can replace CET around PSI without negatively impacting CO2 assimilation. This study revealed that FDPs can effectively support the photosynthetic machinery by maintaining ATP production and stabilizing the photosynthetic apparatus, even when CET pathways are compromised. This substitution ensures that CO2 fixation continues efficiently, thereby supporting overall plant growth and yield. The findings of Wada and colleagues emphasize the versatility and importance of FDPs in maintaining robust photosynthetic performance and plant resilience. The primary genes analysed in their study were FLV1 and FLV4, which are responsible for encoding the essential flavodiiron proteins in rice.»
4) L550-609: As Rubisco activase should not be a section of “2.2.1 Rubisco assembly factor”, the section/section title should be edited. Moreover, the recent review for Rubisco activase should be described and cited here:
Qu, Y., Mueller-Cajar, O. and Yamori, W., 2023. Improving plant heat tolerance through modification of Rubisco activase in C3 plants to secure crop yield and food security in a future warming world. Journal of Experimental Botany, 74(2), pp.591-599.
Answer: This has been done. We added the following paragraph:
«The work of Ku, Müller-Kahara, and Yamori (2023) represents a significant advance in the quest to improve plant heat tolerance by modifying Rubisco activase in C3 plants. In plants, C3 Rubisco activase is vital for maintaining Rubisco functionality, especially under stress conditions such as high temperatures. This article delves into the genetic basis of heat tolerance by examining two specific genes, RCA1 and RCA2, which encode different isoforms of Rubisco activase. These isoforms play distinct roles in the regulation and activation of Rubisco, and their functionality is highly sensitive to temperature fluctuations. Researchers emphasize that under high-temperature conditions, the effectiveness of Rubisco activase decreases, which leads to a decrease in photosynthetic capacity and, consequently, yield. To solve this problem, Ku, Müller-Kahar and Yamori proposed genetic modifications that increase the thermostability and activity of Rubisco activase. Their approach involves changing certain amino acids in the Rubisco activase protein to increase its thermostability. Using site-directed mutagenesis and protein engineering techniques, the researchers were able to create modified versions of Rubisco activase with improved heat tolerance”.
5) A section for Aquaporin should added before/after the “2.2.5. Carbonic Anhydrase (CA)”.
Answer: This has been done.
6) Aquaporins are also known as CO2 transporters. It would be good to describe its information here. The most recent review is, for example,
Chen, J., Yue, K., Shen, L., Zheng, C., Zhu, Y., Han, K. and Kai, L., 2023. Aquaporins and CO2 diffusion across the biological membrane. Frontiers in Physiology, 14, p.1205290..
The following papers are well cited:
Groszmann, M., Osborn, H.L. and Evans, J.R., 2017. Carbon dioxide and water transport through plant aquaporins. Plant, Cell & Environment, 40(6), pp.938-961. ;
Kaldenhoff, R., Kai, L. and Uehlein, N., 2014. Aquaporins and membrane diffusion of CO2 in living organisms. Biochimica et Biophysica Acta (BBA)-General Subjects, 1840(5), pp.1592-1595.
Uehlein N, Lovisolo C, Siefritz F, Kaldenhoff R. The tobacco aquaporin NtAQP1 is a membrane CO2 pore with physiological functions. Nature. 2003 Oct 16;425(6959):734-7..
Answer: We agree. We added the following paragraphs: “The understanding of plant aquaporins, especially their role in CO2 transport, has been advanced by several key studies. Chen et al. (2023) provided a comprehensive overview of the role of aquaporins in CO2 diffusion across biological membranes. Their study highlights the complexity of CO2 permeability in membranes and the potential involvement of aquaporins in facilitating CO2 transport. They discuss the challenges of measuring CO2 permeability and suggest that further research is needed to better understand the molecular mechanisms underlying CO2 transport through aquaporins. The authors highlight the importance of aquaporin expression levels and the role of other membrane proteins and sterols in influencing CO2 permeability.
Groszmann et al. (2017) focused on the transport of CO2 and water through plant aquaporins, in particular studying their physiological roles and how environmental conditions influence their functionality. The effects of altered aquaporin expression on mesophyll conductance and photosynthesis have been investigated. This study used transgenic plants to demonstrate that changes in aquaporin levels correlate with changes in mesophyll conductance and CO2 assimilation rates.
Kaldenhoff et al. (2014) studied the diffusion of CO2 through aquaporins and suggested that these proteins are required for the efficient transport of CO2 across membranes. They discuss the limitations of traditional CO2 diffusion models and suggest that aquaporins provide a regulated pathway for CO2 transport that is critical for processes such as photosynthesis and respiration. This study highlights the role of certain aquaporins in various plant species and their potential to increase CO2 permeability.
Uehlein et al. (2003) identified the tobacco aquaporin NtAQP1 as a CO2 membrane pore with significant physiological functions. Their study showed that NtAQP1 facilitates CO2 transport by influencing photosynthesis and water use efficiency in plants. This study was a pioneer in showing that aquaporins can function as gas channels, opening new avenues for understanding the role of these proteins in plant physiology.
Several genes encoding aquaporins have been shown to improve the permeability of plants to CO2. For example, the genes AtPIP1;2, AtPIP1;3, and AtPIP2;6 have been studied in Arabidopsis thaliana, with mixed results regarding their effects on mesophyll conductance and photosynthetic efficiency. Other studies have focused on aquaporins from crops such as rice (OsPIP1;2, OsPIP1;3) and maize (ZmPIP1;5, ZmPIP1;6), as well as barley (HvPIP2;1, HvPIP2;2, HvPIP2;3, HvPIP2;5) and Italian millet (SiPIP2;7). These studies generally support a role for aquaporins in improving CO2 transport, although the results may vary depending on the growth conditions and the specific aquaporin studied.
The oligomeric and phosphorylated states of aquaporins also influence their functionality, affecting their ability to transport CO2. For example, phosphorylation can alter the pore size and transport capacity of aquaporins, affecting their effectiveness under different environmental conditions.
The overexpression of aquaporins enhances the ability of plants to cope with various abiotic stresses, including salinity, extreme temperature, and drought. Increasing the expression of aquaporins is a promising strategy for improving the photosynthetic performance and water use efficiency of plants, especially under drought stress [123].
7) The section for “3. High-Temperature, Light, Drought, and Salinity Stress Limiting Photosynthesis“ is not well summarized in the current manuscript. As the content of this section is not enough, I think that it would be good to delete this section. Instead, it might be good to point it out briefly in “4. Conclusion”.
For example, plants in natural environments must cope with diverse, highly dynamic, and unpredictable conditions (Yamori, 2016, Journal of Plant Research 129:379–395). The effects of global warming on plant growth and yield have attracted considerable attention (for a review, see Yamori et al. (2014) Photosynthesis Research 119, 101-117). [Then, you can briefly insist on the consideration for strategies for enhancing photosynthesis.]
Answer: We proposed a compromise: instead of removing the sections on stress, we will expand and enhance them to emphasize the critical importance of improving plant adaptation to various stress conditions.
8) Minor points
1) L46-47: The description is wrong, since Light dependent reactions via thioredoxins are included in the Calvin cycle. Please edit this sentence.
Answer: We agree. We added this paragraph.
“Light-independent reactions, also known as the photosynthetic carbon reduction cycle or the Calvin cycle, take place in the stroma of the chloroplast and result in the fixation of CO2, and they include light-dependent reactions via thioredoxins [1,2].
2) L71, L298-299: please cite the appropriate paper (Yamori et al. 2016 Plant, Cell & Environment 39, 80-87.)
Answer: This has been done.
Minor editing of English language required
Answer: This has been done. We improved the English throughout the text.
Reviewer 2 Report
Comments and Suggestions for Authors
The manuscript by Nazari et al. entitled “Enhancing photosynthesis and plant productivity through genetic modification”, presents a very good review on transgenic approaches for improving the photosynthesis efficiency in plants, thereby potentially enhancing the crop productivities. It is well written and summarizes the established work covering most of the important genes involved in the major photosynthetic pathways in C3 and C4 plants.
The major comment is that the innovations carried by the crassulacean acid metabolism (CAM) plants. It’s known that around 90% of the higher plants perform C3 photosynthesis, and about 3% of the rest are C4 plants. The CAM plants occurs in roughly 6% and similar to C4 plants, have evolved multiple times. Both C4 and CAM plants enrich carbon dioxide molecules around the RUBISCO for carbon assimilation reactions, but whereas the C4 plants have the CO2 concentration mechanism (CCM) by spatially separate the Calvin cycle and electron transfer chains of the photosynthetic routes, the CAM plants utilize the CCM by temporally separate the photosynthetic routes. The genomes and important genes involved in improving the water use efficiency in CAM plants have been well studied. For example, the pineapple (Ming et al. Nature Genetics 2015, 47, 1435) and Kalanchoe (Yang et al. Nature Communications 2017, 8, 1899), as well as the rewiring and positive selection of genes in the Agave (Yin et al. BMC genomics 2018, 19, 588). There are also transgenic studies in CAM plants. For example, overexpression of the PEPC gene (which had been carefully discussed in present manuscript) in Agave significantly improves the plant growth, too (Liu et al. Cells 2021, 10, 582). In another study, by introducing a transcription factor from the CAM plant Agave to the C3 plant Populus can increase the biomass yield by 166% (Liu et al. Plant Physiology 2023, 191, 1492).
Because the CAM pathway represents a natural innovation of the plant to enhance the photosynthesis and productivity in response to lower CO2 concentration and stress for water shortage, it is important to add discussions to the CAM photosynthesis. FYI, there was a very good perspective on the CAM plant: Yang et al. New Phytologist 2015, 207, 491.
Author Response
The manuscript by Nazari et al. entitled “Enhancing photosynthesis and plant productivity through genetic modification”, presents a very good review on transgenic approaches for improving the photosynthesis efficiency in plants, thereby potentially enhancing the crop productivities. It is well written and summarizes the established work covering most of the important genes involved in the major photosynthetic pathways in C3 and C4 plants.
The major comment is that the innovations carried by the crassulacean acid metabolism (CAM) plants. Approximately 90% of higher plants perform C3 photosynthesis, and approximately 3% of the remaining plants are C4 plants. Approximately 6% of CAM-tolerant plants, similar to C4 plants, have evolved multiple times. Both C4 and CAM plants enrich carbon dioxide molecules around the RUBISCO for carbon assimilation reactions, but whereas the C4 plants have the CO2 concentration mechanism (CCM) by spatially separating the Calvin cycle and electron transfer chains of the photosynthetic routes, the CAM plants utilize the CCM by temporally separating the photosynthetic routes. The genomes and important genes involved in improving water use efficiency in CAM plants have been well studied. For example, pineapple (Ming R, VanBuren R, Wai CM, Tang H, Schatz MC, Bowers JE, Lyons E, Wang ML, Chen J, Biggers E, Zhang J. The pineapple genome and the evolution of CAM photosynthesis. Nature genetics. 2015 Dec;47(12):1435-42.) and Kalanchoe (Yang X, Hu R, Yin H, Jenkins J, Shu S, Tang H, Liu D, Weighill DA, Cheol Yim W, Ha J, Heyduk K. The Kalanchoë genome provides insights into convergent evolution and building blocks of crassulacean acid metabolism. Nature communications. 2017 Dec 1;8(1):1899. ), as well as the rewiring and positive selection of genes in Agave (Yin, H., Guo, H.B., Weston, D.J., Borland, A.M., Ranjan, P., Abraham, P.E., Jawdy, S.S., Wachira, J., Tuskan, G.A., Tschaplinski, T.J. and Wullschleger, S.D., 2018. Diel rewiring and positive selection of ancient plant proteins enabled the evolution of CAM photosynthesis in Agave. BMC genomics, 19, pp.1-16.). There are also transgenic studies in CAM plants. For example, overexpression of the PEPC gene (which was carefully discussed in the present manuscript) in Agave significantly improved plant growth (Liu, D., Hu, R., Zhang, J., Guo, H.B., Cheng, H., Li, L., Borland, A.M., Qin, H., Chen, J.G., Muchero, W. and Tuskan, G.A., 2021). Overexpression of an agave phospho enol pyruvate carboxylase improves plant growth and stress tolerance. Cells, 10(3), p.582.). In another study, the introduction of a transcription factor from the CAM plant Agave to the C3 plant Populus increased the biomass yield by 166% (Liu, D., Tang, D., Xie, M., Zhang, J., Zhai, L., Mao, J., Luo, C., Lipzen, A., Zhang, Y., Savage, E. and Yuan, G., 2023. Agave REVEILLE1 regulates the onset and release of seasonal dormancy in Populus. Plant physiology, 191(3), pp.1492-1504. ).
Because the CAM pathway represents a natural innovation of plants to enhance photosynthesis and productivity in response to lower CO2 concentrations and stress due to water shortage, it is important to discuss CAM photosynthesis. Yang, X., Cushman, J.C., Borland, A.M., Edwards, E.J., Wullschleger, S.D., Tuskan, G.A., Owen, N.A., Griffiths, H., Smith, J.A.C., De Paoli, H.C. and Weston, D.J., 2015. A roadmap for research on crassulacean acid metabolism (CAM) to enhance sustainable food and bioenergy production in a hotter, drier world. New Phytologist, 207(3), pp.491-504.
Answer: We agree. We have added paragraphs and references.
« The study of crassulacean acid metabolism (CAM) photosynthesis has attracted significant interest due to its water-use efficiency. Approximately 90% of higher plants are known to carry out C3 photosynthesis, and approximately 3% of the remaining plants are C4 plants. Approximately 6% of CAM plants, similar to C4 plants, have evolved several times. C4 and CAM plants enrich carbon di-oxide molecules around RUBISCO for carbon assimilation reactions, but while C4 plants have a CO2 concentration mechanism (CCM), spatially separating the Calvin cycle and electron transport chains of the photosynthetic pathways, CAM plants use CCM to temporarily separate the photosynthetic pathways. The genomes and important genes involved in improving water use efficiency in CAM plants have been well studied.
For example, a study by VanBuren et al. (2015) sequenced the pineapple genome, revealing the genetic basis of CAM photosynthesis. Researchers have identified several key genes, including phosphoenolpyruvate carboxylase (PEPC), phosphoenolpyruvate carboxykinase (PEPCK), malate dehydrogenase (MDH), NADP-malic enzyme (NADP-ME), and pyruvate phosphate dikinase (PPDK). These genes are critical for fixing COâ‚‚ at night and releasing it during the day, allowing efficient water use in dry conditions (VanBuren et al., 2015). Yang et al. (2017) examined the Kalanchoe genome and identified similar key genes involved in CAM photosynthesis. This study highlighted the roles of PEPC, MDH, NADP-ME, starch synthase (SS), starch phosphorylase (SP) and various transport proteins. These results highlight the convergent evolution of CAM, where different plant species have independently evolved similar genetic solutions to thrive in arid environments (Yang et al., 2017). Yin et al. (2018) examined the agave genome and identified key genes and proteins that have undergone daily regulatory changes and positive selection to promote the evolution of CAM photosynthesis. The important genes included PEPC, NAD-malic enzyme (NAD-ME), NADP-malic enzyme (NADP-ME), starch synthase (SS), starch phosphorylase (SP), and circadian clock genes. This study showed how ancient plant proteins were coopted and selectively enhanced to support CAM, highlighting the adaptive significance of genetic rearrangement in response to environmental pressures (Yin et al., 2018).
Transgenic research is also being carried out on CAM plants. For example, overexpression of the PEPC gene in agave also significantly improved plant growth (Liu et al., 2021). In another study, introducing a transcription factor from the CAM plant Agave into a C3 Populus plant increased the biomass yield by 166% (Liu et al., 2023). Because the CAM pathway is a natural plant innovation for improving photosynthesis and productivity in response to lower CO2 concentrations and water stress, it is important to add a discussion of CAM photosynthesis. Yang et al. (2015) developed a roadmap for CAM metabolic research to improve sustainable food and bioenergy production in a hotter, drier world.
Comparative analyses of CAM photosynthesis in pineapple, Kalanchoe, and agave reveal both common genetic mechanisms and unique adaptations. All studies identified PEPC as a critical enzyme for nocturnal COâ‚‚ fixation, highlighting its universal importance in CAM photosynthesis. Genes involved in carbohydrate storage and mobilization, such as starch synthase (SS) and starch phosphorylase (SP), have been identified in various CAM plants. Regulatory networks and circadian clock genes play important roles in optimizing CAM activity according to circadian cycles. However, research has also revealed differences. A study of the pineapple genome (VanBuren et al.) highlighted the role of PEPCK in the diurnal decarboxylation process, whereas this enzyme was less prominent in studies of agave and Kalanchoe. The study of agave (Yin et al.) clearly revealed positive selection and diurnal rearrangement of ancient plant proteins, which was not the subject of studies of pineapple and Kalanchoe. Transport proteins were highlighted in a study by Kalanchoe (Yang et al.), emphasizing the movement of organic acids and metabolites between cellular compartments.
Enhanced CO2 fixation by PEPC may also help plants better tolerate environmental stresses such as high light intensity and temperature fluctuations by optimizing the photosynthetic process. Although the overexpression of C4 enzymes such as PEPC in C3 plants has potential, several challenges remain. It is critical to ensure that the insertion of PEPC does not disrupt the overall metabolic balance of the plant. The integration of C4 pathways into C3 plants requires careful regulation to avoid negative effects on plant growth and development. To fully realize the benefits of C4 photosynthesis in C3 plants, a combination of genetic modifications may be needed, including the insertion of other C4-specific enzymes and regulatory elements [112].»
Reviewer 3 Report
Comments and Suggestions for Authors
The review covers all important aspects of the photosynthetic apparatus and the possibilities to improve the photosynthesis for better crop production.
However, several problems occur:
1. The authors need to structure the review. I would suggest from leave structure, to cell and finally the RNA as adjusting screws to improve photosynthesis. At the moment it is a bit mixed. The part with the photosynthesis should be structured in order of the electron flow. Furthermore, the small RNAs are mentioned in the introduction and later in the text again with very similar content. Also for the structure please describe firstly the TF , afterwards the miRNAs
2. The authors should check the manuscript regarding term utilization. E.g. Calvin Cycle, Calvin Benson cycle, CB cycle. See also CO2 fixation, Rieske protein etc.. And all term should be introduced (in brackets) before an abbreviation is used.
3. The abstract needs to be structured better and the first sentence should be changed to something more introducing the topic.
4. Additionally there are some real mistakes in the text
a. Line 54: most successful genes? What are successful genes?
b. Line 62: FNR is no mechanism
c. Line 69: FNR is no approach
d. Line 139: this sentence needs to be changed, it is meaningless and wrong.
e. Line 167: carotenoids do not need to be protected from light
f. Line 269: the overexpression does not increase the higher quenching.
g. Line 346: I don’t think this is a final version of this paragraph
h. Line 420: 25% of CO2 fixation? I don’t think that is correct
i. Line 448: which enzymes?
j. Line 489: no NH3 was fixed
k. Line 524: the study is not mentioned
l. Line 830: photosynthesis or the membrane or the cell structure or proteins in general? Be more specific
Comments on the Quality of English LanguageCells review photosynthesis
The review covers all important aspects of the photosynthetic apparatus and the possibilities to improve the photosynthesis for better crop production.
However, several problems occur:
1. The authors need to structure the review. I would suggest from leave structure, to cell and finally the RNA as adjusting screws to improve photosynthesis. At the moment it is a bit mixed. The part with the photosynthesis should be structured in order of the electron flow. Furthermore, the small RNAs are mentioned in the introduction and later in the text again with very similar content. Also for the structure please describe firstly the TF , afterwards the miRNAs
2. The authors should check the manuscript regarding term utilization. E.g. Calvin Cycle, Calvin Benson cycle, CB cycle. See also CO2 fixation, Rieske protein etc.. And all term should be introduced (in brackets) before an abbreviation is used.
3. The abstract needs to be structured better and the first sentence should be changed to something more introducing the topic.
4. Additionally there are some real mistakes in the text
a. Line 54: most successful genes? What are successful genes?
b. Line 62: FNR is no mechanism
c. Line 69: FNR is no approach
d. Line 139: this sentence needs to be changed, it is meaningless and wrong.
e. Line 167: carotenoids do not need to be protected from light
f. Line 269: the overexpression does not increase the higher quenching.
g. Line 346: I don’t think this is a final version of this paragraph
h. Line 420: 25% of CO2 fixation? I don’t think that is correct
i. Line 448: which enzymes?
j. Line 489: no NH3 was fixed
k. Line 524: the study is not mentioned
l. Line 830: photosynthesis or the membrane or the cell structure or proteins in general? Be more specific
Author Response
The authors sincerely thank the distinguished reviewers for reviewing our data, providing useful recommendations, and providing friendly criticism.
This review covers all important aspects of the photosynthetic apparatus and the possibilities for improving photosynthesis for better crop production.
However, several problems occur:
- The authors need to structure the review. I would suggest from leaf structure, to cell and finally the RNA as adjusting screws to improve photosynthesis. At the moment, it is a bit mixed. The part involved in photosynthesis should be structured in order of electron flow. Furthermore, the small RNAs are mentioned in the introduction and later in the text again with very similar content. Also for the structure please describe first, the TF , afterwards the miRNAs
Answer: We improved the section order.
- Introduction
- Chloroplast component modifications
2.1. Optimization of Light Harvesting and Pigment Content
2.2. Photosystem II
2.3. Cytochrome b6f Complex
2.4. Photosystem I
2.5. Electron Transport Chain
2.6. Chloroplast ATP synthase
- Carbon Assimilation Efficiency
3.1. RuBisCO as a Target to Improve Carbon Assimilation Efficiency
3.2. Rubisco Assembly Factors
3.3. Sedoheptulose-1,7-bisphosphatase (SBPase)
3.4. Genes from C4 Plants
3.5. Photorespiration
- Cellular transport and regulation
4.1. Sucrose Transporters
4.2. Aquaporins
4.3. Carbonic Anhydrase (CA)
- Gene Regulation
5.1. High-Pigment Epistasis 1 (HPE1)
5.2. Transcription Factors (TFs)
5.3. MicroRNAs
- Environmental Stressors Limiting Photosynthesis
6.1. Temperature stress
6.2. Light stress
6.3. Water and Salt Stresses
6.4. Biotic stress
6.5. Cross-adaptation
- Conclusions
- The authors should check the manuscript regarding term utilization. These include the Calvin cycle, the Calvin–Benson cycle, and the CB cycle. See also CO2 fixation, Rieske protein, etc. In addition, all term should be introduced (in brackets) before an abbreviation is used.
Answer: We improved the abbreviations used.
- The abstract needs to be structured better and the first sentence should be changed to something more introducing the topic.
Answer: We improved the abstract, and the first sentence was changed.
- Additionally, there are some real mistakes in the text
- Line 54: most successful genes? What are successful genes?
Answer: We improved “promising genes”.
- Line 62: FNR is no mechanism
Answer: We agree. Another important component is ferredoxin-NADP+ reductase (FNR).
- Line 69: FNR is no approach
Answer: We agree. For example, transgenic Nicotiana tabacum plants overexpressing FNR exhibited enhanced photosynthetic capacity and growth under various light conditions [6]. This modification can improve the overall redox balance and metabolic efficiency in plants.»
- Line 139: this sentence needs to be changed, it is meaningless and wrong.
Answer: We agree. The Calvin cycle involves many genes that are valuable for genetic modification. Additionally, noncoding miRNAs and transcription factors can accelerate cell division, facilitate transitions to necessary ontogenetic stages, and significantly increase stress resistance.»
- Line 167: carotenoids do not need to be protected from light
Answer: We agree. “The expression of LCYB in different plants has relatively specific effects on increasing the synthesis of carotenoids, which not only perform their main function but also protect against various stress factors, including excess light”.
- Line 269: the overexpression does not increase the higher quenching.
Answer: We agree and have removed this term.
- Line 346: I don’t think this is a final version of this paragraph
Answer: We agree. It is done.
- Line 420: 25% of CO2 fixation? I don’t think that is correct
Answer: We agree and have removed this term.
- Line 448: which enzymes?
Answer: We agree. “Pathway 3 involves the introduction of two enzymes, malate synthase and malate dehydrogenase, leading to the formation of reducing equivalents and shifting CO2 release reactions from mitochondria to chloroplasts. Preliminary results in transgenic A. thaliana indicate increased photosynthesis and biomass via this pathway Shi, X., & Bloom, A. (2021). Photorespiration: the futile cycle?. Plants, 10(5), 908. [81].»
- Line 489: no NH3 was fixed
Answer: We agree and have removed this term.
- Line 524: the study is not mentioned
Answer: This has been done.
- Line 830: photosynthesis or the membrane or the cell structure or proteins in general? Be more specific
Answer: It is done «Photosynthesis is highly susceptible to heat stress, which causes significant damage to various components of the photosynthetic machinery and leads to cellular energy imbalances [137]. High temperatures disrupt the thylakoid membrane, damaging membrane-associated electron carriers and enzymes [138-140]. Specifically, heat stress harms Photosystem I (PSI), Photosystem II (PSII), the cytochrome b6f complex (Cyt b6f), and Rubisco [138]. The increase in thylakoid membrane fluidity under thermal stress compromises the light-harvesting complex and disrupts the structural integrity of PSII [141].»
Round 2
Reviewer 2 Report
Comments and Suggestions for Authors
The authors answered the suggestions and questions previously asked. I have no more comments for this revision.
Author Response
We are grateful to the reviewer for his time
Reviewer 3 Report
Comments and Suggestions for Authors
Introduction still needs to be shortened, to avoid repetition with the chapters. Please just mention the different target points: 1. Photosynthesis 2. Calvin cycle 3 cellular processes 4. TF and miRNA etc.
Line 79-94 switch paragraph to keep an order
Line 796 the findings of QU represent
Line 874 Cam photosynthesis has …
Line 901 Research with transgenic CAM plants has been carried out, for example…
Line 1043 Groszman investigated the role of aquaporins in…
Line 1049 Kaldenhoff discovered that aquaporins are involved in CO2 transport in plant membranes
Line 1072 reference is missing
Line 1245 paragraph can be shortened and avoid repetition
Line 1339 paragraph has to be checked for prepetition and can be shortened
Line 1355 photosynthesis in inhibited by low temperature as …
Line 1402ff avoid repetition
Comments on the Quality of English Language
overall the english needs to be checked, several sentensences are wrong, see also the review. But i did not check the whole manuscript, since i am not a lector.
Author Response
- Introduction still needs to be shortened, to avoid repetition with the chapters. Please just mention the different target points: 1. Photosynthesis 2. Calvin cycle 3 cellular processes 4. TF and miRNA etc.
Answer: For each paragraph we left 3-5 sentences, shortened according to the Reviewer's recommendation.
- Line 79-94 switch paragraph to keep an order
Answer: It is done.
- Line 796 the findings of QU represent
Answer: It is done.
- Line 874 Cam photosynthesis has …
Answer: It is done.
- Line 901 Research with transgenic CAM plants has been carried out, for example…
Answer: It is done.
- Line 1043 Groszman investigated the role of aquaporins in…
Answer: It is done.
- Line 1049 Kaldenhoff discovered that aquaporins are involved in CO2 transport in plant membranes
Answer: It is done.
- Line 1072 reference is missing
Answer: It is done. ( Groszmann · 2017)
- Line 1245 paragraph can be shortened and avoid repetition
Answer: It is done.
- Line 1339 paragraph has to be checked for prepetition and can be shortened
Answer: It is done.
- Line 1355 photosynthesis is inhibited by low temperature as …
Answer: It is done.
- Line 1402ff avoid repetition
Answer: It is done.
- overall the english needs to be checked
Answer: It is done.